# Continuous evolution of a halogenase enzyme with improved solubility and activity for sustainable bioproduction

Andre Arashiro Pulschen[1,3], Justin Booth [1,2,3], Ari Satanowski [1,2], Christelle Soudy [1], Joaquin Caro-Astorga [1], Osaid Ather [1,2], Namita Patel[1], Ali Alidoust[1], Samir Aoudjane[1], Lily Nematollahi[1,2] & Erika DeBenedictis [1,2] ✉

Halogenation enhances the stability and function of pharmaceuticals, biomaterials, and industrial compounds. However, chemical halogenation of molecules and peptides can lack stereoselectivity and require the use of toxic chemicals. Although enzymatic halogenation can improve selectivity and reduce environmental impact, current halogenases are inefficient and insoluble, leading to low yields that limit their applications. Here, we develop RebH$_{Evo4}$, a soluble and highly active tryptophan halogenase, containing 12 mutations that confer 37-fold and 44-fold increases in 7-chloro- and 7-bromotryptophan production respectively, in vivo. To create RebH$_{Evo4}$, we devise an aminoacyl-tRNA synthetase-based halogenase biosensor and conduct over 500 hours of phage-assisted continuous evolution (PACE). Use of RebH$_{Evo4}$ in a bioreactor results in the production of 2.7 g/L of halogenated tryptophan. When coupled with a downstream enzyme, RebH$_{Evo4}$ allows 36-fold increased yields of halogenated tryptamines compared to the wild-type enzyme. Additionally, RebH$_{Evo4}$ enables efficient production of genetically encoded antimicrobial halogenated peptides. The efficient, site-specific halogenation enabled by our evolved halogenase will accelerate sustainable biomanufacturing of halogenated drugs.

Halogenated compounds are central to modern medicine, with about 25% of approved drugs and over 80% of agrochemicals incorporating halogens to enhance bioavailability and stability[1–3]. While chemical halogenation is a well-established method, it often requires harsh conditions involving toxic and corrosive reagents[4,5] and can lack regio- and enantioselective control. Enzymatic halogenation is emerging as a powerful, regioselective, and environmentally sustainable alternative[5]. Tryptophan halogenases such as RebH (which can natively halogenate the tryptophan indole group at position 7 with both chloride and bromide) are among the best-studied biocatalysts for halogenation. These enzymes rely on a two-step catalysis, first oxidising a flavin cofactor to drive production of an intermediate hypohalous acid

(HOX) species, then ensuring precise electrophilic halogenation at the indole ring of tryptophan and related compounds[6].

Tryptophan and its derivatives are key precursors of several molecules of interest, including those with applications in the food industry, pigments[7,8], agriculture, and medicine[9–12]. Halogenation of Tryptophan or indole is necessary for the production of important molecules such as the pigment tyrian purple[7], and the antifungal pyrrolnitrin[11]. Additionally, Tryptophan halogenation has been explored for the diversification of natural products, which can improve stability, pharmacokinetics, and activity[5]. It has also been explored for the diversification of other active molecules, like violacein[8] and plant monoterpenes, such as alstonines and serpentines[13], carbolines and

[1]The Francis Crick Institute, London, UK. [2]The Biodesign Laboratory, London, UK. [3]These authors contributed equally: Andre Arashiro Pulschen, Justin Booth. ✉e-mail: erika@biodesignlab.co.uk

pro-drugs halogenated kynurenines[12]. Selective incorporation of halogenated tryptophan can also improve protein and peptide drugs. For example, a recently discovered halogenated variant of the antibiotic Darobactin A is more active than its non-halogenated compound[14]. Krisynomycin and Nisin are additional examples of molecules in which their activity and stability are improved due to halogenation[15,16].

Previous efforts have managed to reach gram-scale yields using flavin-dependent halogenases in vitro, through an 8-day reaction with a cross-linked aggregate of 3 different purified enzymes (ADH, PrnF, and RebH)[17]. However, to date, there has been limited success in using flavin-dependent halogenases to produce substantial amounts of halogenated tryptophan compounds in vivo. This is due to poor solubility[7], low enzymatic activity, substrate inhibition[18], and temperature sensitivity[19], leading to low fermentation yields[19] and slow reaction rates[7,12]. Although efforts have been made to engineer halogenases with expanded substrate scope[20,21], enhanced thermo-tolerance in in vitro conditions[22], and decreased leakiness of intermediate HOX[23], it remains a challenge to obtain highly active variants. As a consequence, despite their limitations, wild-type sequences continue to comprise the majority of enzymes used for in vivo bioproduction and fermentation[7,8,12,13,24]. Notably, there are no described variants with improved solubility and improved in vivo activity; instead, fusion with more soluble proteins, such as maltose-binding protein and flavin reductases, and co-expression of chaperones are used in an attempt to overcome this[7,25,26].

In this study, we sought to overcome these challenges by applying continuous directed evolution to the flavin-dependent tryptophan halogenase RebH. To establish a readout for halogenation activity, we developed a biosensor based on the engineered aminoacyl-tRNA synthetase (aaRS) ChPheRS-4[27], coupling the biosynthesis of the halogenated non-canonical amino acid (ncAA) 7-chlorotryptophan (7-Cl-Trp) to expression of the sfGFP gene. We then adapted our biosensor to link the activity of RebH to phage propagation, allowing us to conduct over 500 h of Phage-Assisted Continuous Evolution (PACE). Our final evolved variant, RebH$_{Evo4}$, contains 12 mutations and exhibits improved solubility and activity compared to the wild-type enzyme. When coupled with the downstream enzyme RgnTDC, it allowed the efficient production of halogenated tryptamines. Additionally, we used RebH$_{Evo4}$ for bioproduction of halogenated antimicrobial peptides (AMPs), demonstrating its use for the efficient manufacture of proteins with site-specific halogenation in *Escherichia coli*.

## Results

### Aminoacyl-tRNA synthetase biosensor to detect halogenation in vivo

We envisioned that an aaRS that is selective for a halogenated ncAA could function as a biosensor, enabling amber stop codon (TAG) suppression only when a halogenated amino acid is added to the media, or halogenation occurs in vivo. We designed a genetic circuit to utilise the engineered chimeric aaRS ChPheRS-4, which has previously been reported to incorporate 7-Cl-Trp into proteins at TAG codons[27] (Fig. 1a). By incorporating a TAG amber stop codon within the gene encoding a green fluorescent protein (sfGFP), using a GFP-151-TAG cassette, amber suppression using the halogenated ncAA allows full sfGFP translation and thus fluorescence, which can be used as an indirect measure of tryptophan halogenation. In our hands, using a BW25113 ΔtnaA (tryptophanase deletion) strain to prevent tryptophan degradation, ChPheRS-4 demonstrated weak amber suppression, as measured by GFP signal (Supplementary Fig. 1A). To improve the biosensor, we searched the literature for variants of the aaRS-tRNA pair that have been shown to improve amber suppression on related ncAAs and identified two possible variants: the aaRS mutation S333C and the tRNA mutation 3C11[28]. Testing these variants confirmed improvements in amber suppression and thus 7-Cl-Trp incorporation into sfGFP

(Supplementary Fig. 1A, B). As media composition is known to strongly impact the performance of amber suppression circuits, we compared amber suppression in five different types of media (Fig. 1b). The largest fold-change occurred using DRM media, which was originally formulated for PACE experiments[29], while the largest absorbance-normalised GFP signal was observed when using M9. We chose to combine both types of media in a 1:9 DRM/M9 ratio, which allowed strong GFP signal, with very low leakiness (Fig. 1c). The resulting biosensor enables robust detection of 7-Cl-Trp at concentrations as low as 1 μM (ninefold increased signal over background) and with the operational range reaching as high as 125 μM to 250 μM (95-fold higher signal over background), at 30 °C (Fig. 1c).

We observed that GFP signal was substantially reduced when the media was supplemented with canonical L-tryptophan in addition to 7-Cl-Trp (Fig. 1c), indicating that non-halogenated tryptophan acts as an inhibitor of the ChPheRS-4 synthetase and, in effect, dramatically increases the amount of 7-Cl-Trp required for maximal expression. Additionally, sfGFP signal was not observed when 7-Cl-Trp was replaced with tryptophan, indicating that the aaRS does not promiscuously charge its tRNAs with a non-halogenated substrate. These observations indicate that negative selection against incorporation of canonical tryptophan is not necessary for this aaRS, and that the stringency of our biosensor can be adjusted with the simple addition of canonical tryptophan to the media.

We further characterised our biosensor by measuring its activity at different temperatures. In previous work, amber suppression with ChPheRS-4 has been measured at 30 °C[27]. In our hands, we found that the efficiency of amber suppression is reduced at 34 °C and 37 °C when compared to 30 °C (Fig. 1d). However, no GFP signal was detected in any of the tested temperatures in the absence of the ncAA, confirming that overall synthetase fidelity is maintained throughout this temperature range (Fig. 1d).

Moving on from exogenous addition of 7-Cl-Trp, we next sought to confirm that our biosensor could be used to measure the activity of halogenase enzymes expressed in vivo. Since all known tryptophan halogenases rely on an available pool of reduced flavin cofactors, co-expression with a flavin reductase has been shown to improve in vivo yields of halogenated tryptophan[7,30]. We co-expressed the flavin-dependent halogenase RebH (*Lechevalieria aerocolonigenes*, Uniprot Q8KHZ8) with the flavin reductase Fre (Uniprot P0AEN1) in *E. coli* (Fig. 1a), and found that expression of the halogenase inside the cell results in a GFP signal that is similar in magnitude to direct addition of 7-Cl-Trp to the media, suggesting that the halogenase was actively producing this halogenated ncAA (Fig. 1e). To further confirm that the GFP production is due to the presence of RebH and not the flavin reductase, we tested a construct replacing RebH with the innocuous maltose binding protein (MBP), and confirmed that no GFP production was observed (Fig. 1e). Together, these data demonstrate that the biosensor enables robust measurement of halogenase activity and that 7-Cl-Trp biosynthesis occurs in vivo.

### Halogenation-dependent phage propagation

Next, we sought to adapt the GFP readout of our biosensor to an M13 phage readout, a prerequisite for conducting halogenase evolution using PACE. PACE[31] is a continuous evolution method in which the protein to be evolved is encoded in place of the pIII-encoding gene in an M13 bacteriophage. The resulting loss of pIII protein production prevents replication of the phage unless pIII is produced by a circuit encoded in the *E. coli* cell it infects. Since circuit pIII output is constructed to be dependent on the activity of the protein of interest, phage propagation is linked to the function of the protein being evolved. When propagated against a constantly-replaced pool of host cells under high mutagenesis, PACE allows for rapid exploration of the sequence-function landscape and selection of activity-improving mutations[29].

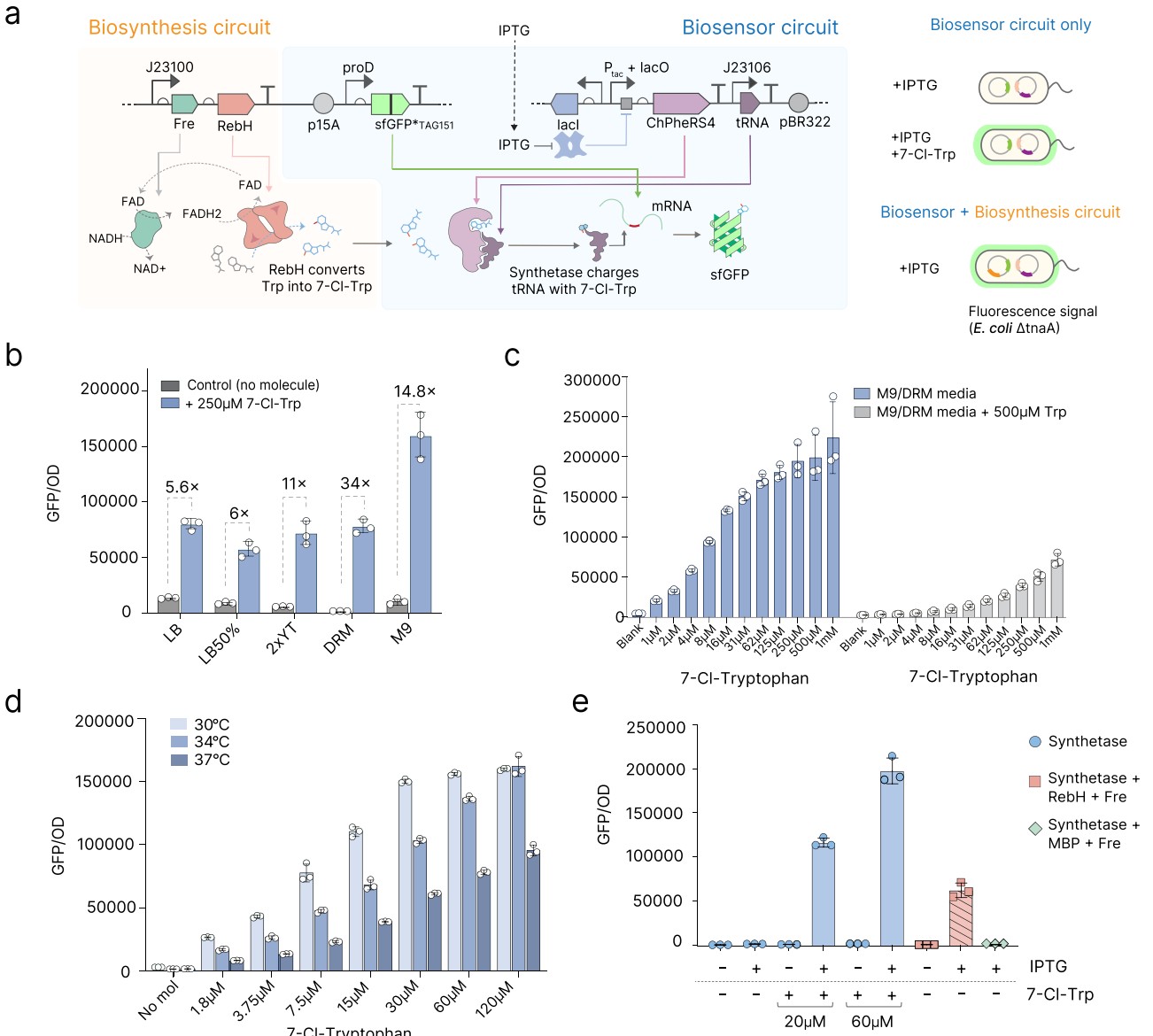

**Fig. 1 | Generation of an AARS-biosensor and biosynthesis circuit for halogenation. a** Schematic of our aaRS-based biosensor for 7-Cl-Trp, and a circuit for biosynthesis of 7-Cl-Trp using the flavin-dependent halogenase RebH that results in amber codon suppression, incorporation of 7-Cl-Trp, and thus full-length expression of sfGFP. The biosensor can be tested independently or combined with the biosynthesis pathway in the same *E. coli* cell. **b** Measurement of biosensor readout in different media backgrounds. The graph depicts GFP normalized by $OD_{600}$. Fold-change between the averages are shown. Error bars show mean and SD between 3 biological replicates. **c** Measurement of the biosensor readout in a 1:9 DRM/M9 media background, with or without supplementation with canonical tryptophan.

Error bars show mean and SD between 3 biological replicates. **d** Measurement of the biosensor readout at 30 °C, 34 °C, and 37 °C, with different concentrations of 7-Cl-Trp. Error bars show mean and SD between 3 biological replicates. **e** Measurements of the biosensor and biosynthesis circuit within the same cell, at 34 °C. Bars indicate the use of the synthetase (ChPheRS-4), Halogenase (RebH), Flavin reductase (Fre), and Maltose Binding Protein (MBP). Error bars show mean and SD between 3 biological replicates. For all GFP measurements, reads were normalized by final $OD_{600}$, to ensure that changes in growth rate do not impact the accuracy of measurements. Source data for this figure is available in Source data.

For our circuit, we expressed RebH on the phage under the strong constitutive proK promoter, while the aaRS-tRNA pair and flavin reductase are encoded by accessory plasmids. The sfGFP*TAG151 from our previous circuit was replaced with pIII*TAG29, which has previously been used to tie amber suppression to M13 phage propagation[32]. Upon infection of *E. coli* host cells (ΔtnaA cells conjugated with the F-plasmid), RebH is expressed from the phage and induces amber suppression by production and translational incorporation of 7-Cl-Trp, thereby allowing production of pIII and phage propagation (Fig. 2a). We tested this design using phage plaquing and confirmed that RebH-encoding phage can successfully propagate without the addition of ncAA to the media, whereas a negative

control empty phage (lacking RebH) could only propagate when 7-Cl-Trp was added to the media (Fig. 2b). Since RebH is known to be a highly temperature-sensitive enzyme, we next probed the temperature-dependence of this system. At 34 °C, phage propagation was very limited, but still detectable for the phage encoding RebH, whereas an empty phage suffered from strong de-enrichment (Fig. 2c). Finally, phage propagation at 37 °C was only observed upon addition of 7-Cl-Trp to the media (Fig. 2c). These data indicate that all components of the system are sufficiently active at 34 °C, with a maximum of three-log-fold phage enrichment upon sufficient 7-Cl-Trp presence (Fig. 2c), making it a viable starting temperature for evolution.

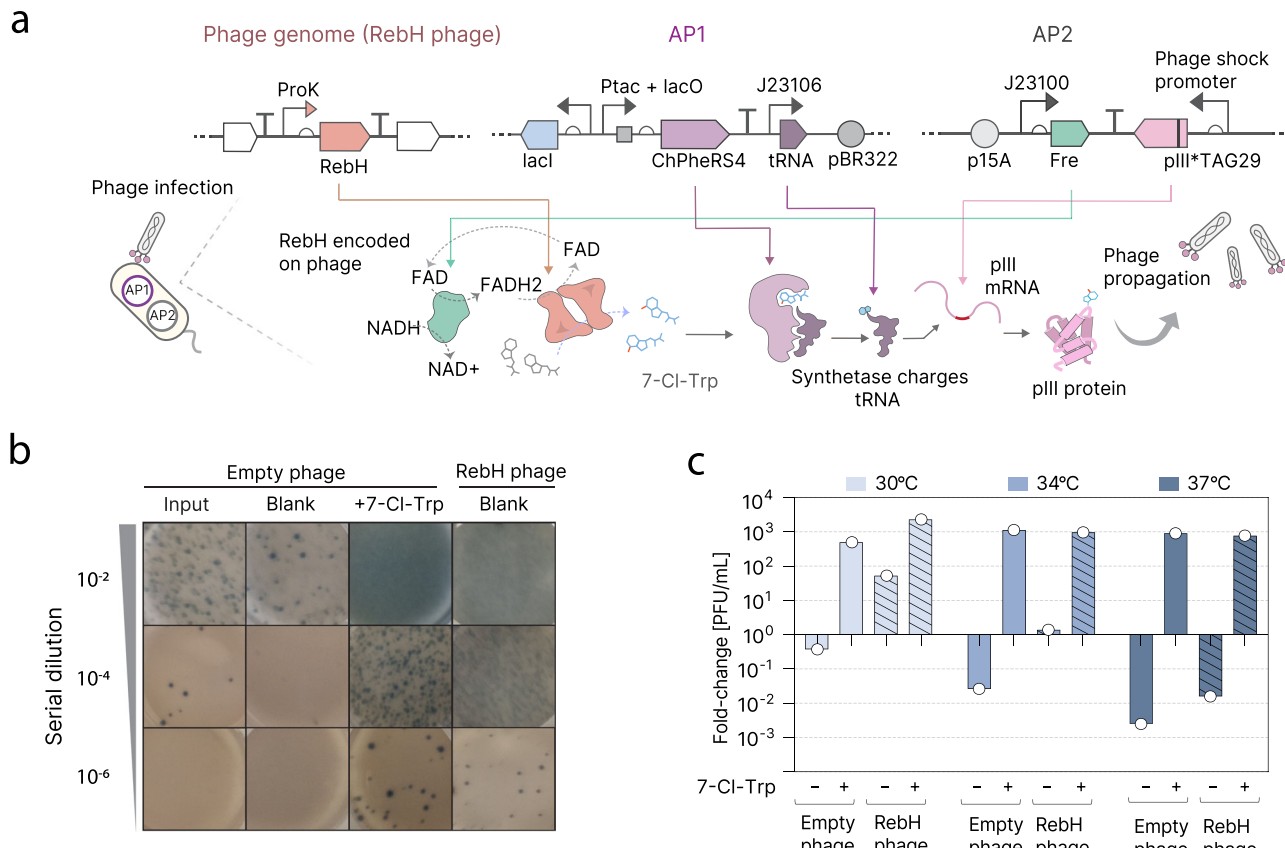

**Fig. 2 | Halogenase M13 phage-based gene circuit. a** M13 phage-based halogenase circuit. RebH gene is encoded on the phage; Accessory Plasmid (AP) 1 produces aaRS and tRNA, AP2 produces Fre and pIII mRNA containing a single amber strop codon at position 29. Only 7-Cl-Trp production allows amber suppression of the pIII-TAG29 and thus phage propagation. **b** Plaque results of an overnight phage enrichment at 30 °C performed in liquid DRM:M9 media in the presence of phage bearing RebH or no gene, and with either no small molecule added or direct addition of 7-Cl-Trp. **c** Overnight phage enrichment (in DRM:M9 media, using BW25113 ΔtnaA F⁺) across different temperatures and in the presence or absence of phage-encoded RebH or supplementation with 7-Cl-Trp (200 μM). PFU/mL stands for plaque-forming units per mL. Source data for this figure is available in Source data.

## Phage-assisted continuous evolution of RebH

We evolved RebH using PACE for a total of 560 h of continuous flow divided over three phases (Fig. 3a). In the first phase, we evolved WT RebH with continuous in vivo mutagenesis induced by the mutagenesis plasmid MP6[33] which expresses a range of error-prone replication machinery for 240 h (Fig. 3b), gradually increasing the temperature from 34 to 37 °C to push the selection to more thermostable variants. Sequencing 24 clonal phage from the final timepoint, we observed 2 co-occurring mutations that were fixed (present in more than 50% of the sequenced phages): V256I and T385I (Supplementary Fig. 2A), suggesting that these are the most beneficial mutations for the enzyme activity. We also found additional mutations (M430L, T348A, L188F, and L380F) that did not reach fixation. However, since both T348A and M430L co-occurred with the fixed V256I and T385I, we combined these mutations and tested, using our GFP biosensor circuit (Supplementary Fig. 2B) and obtained a further improved variant with 4 mutations (256I, 348A, 385I, 430L), which we called RebH_Evo1, that demonstrated improved GFP signal (Fig. 3c, d).

The success of combining different mutations observed from the first phase of PACE led us to use a similar approach in our second and third phases of continuous evolution, where we initiated PACE with pre-diversified libraries (Supplementary Fig. 2C), resulting in successful generation of a larger diversity of phage with several mutations relative to initiating evolution with a clonal genotype (Supplementary Fig. 2A).

In our second phase, we templated an error-prone PCR library on the RebH_Evo1 mutant and used this library to initiate continuous

evolution. After 120 h of continuous flow, we analysed the sequence of 40 clonal phages. We could not observe fixed mutations in this round, but five different mutations (T496R, D101N, A16S, A32V and A50T) were enriched (2 or more plaques with the mutation), alongside several mutations observed once (Supplementary Fig. 2A). We attribute such an increase in the number of variable mutations to our use of a diversified phage library. Interestingly, among the sequences, we observed several instances in which independent phage had mutations at very close positions structurally (Fig. 3e), such as D101N and G102S, and T496R and Q494K, which could suggest a hot-spot for mutations. Also, residues close to the flavin binding pocket, such as N326K, were observed. We then decided to test combining the mutations in two steps: initially focusing mostly more on the enriched mutations and N326K (See Supplementary Fig. 2D). From our tested panel, we obtain RebH_Evo2, containing three additional mutations that arose during the second phase (7 mutations total, Fig. 3c). Building on RebH_Evo2, we tested other mutations, including the remaining enriched mutations A32V and D101N, as well other single mutants with close proximity to the flavin binding pocket (L233M), and surface charged mutants, such as Q494K. We then obtained RebH_Evo3, a variant containing 10 mutations relative to RebHWT, that achieved the highest activity of all variants tested. Interestingly, this variant contains 2 residues close to each other (Q494K and T496R), both on the surface and charged residues, which are known to potentially affect solubility[34,35].

Finally, in our third PACE phase, we repeated the process using RebH_Evo3 as the template for library diversification, which was used to initiate evolution. After 200 h of continuous flow, we obtained one

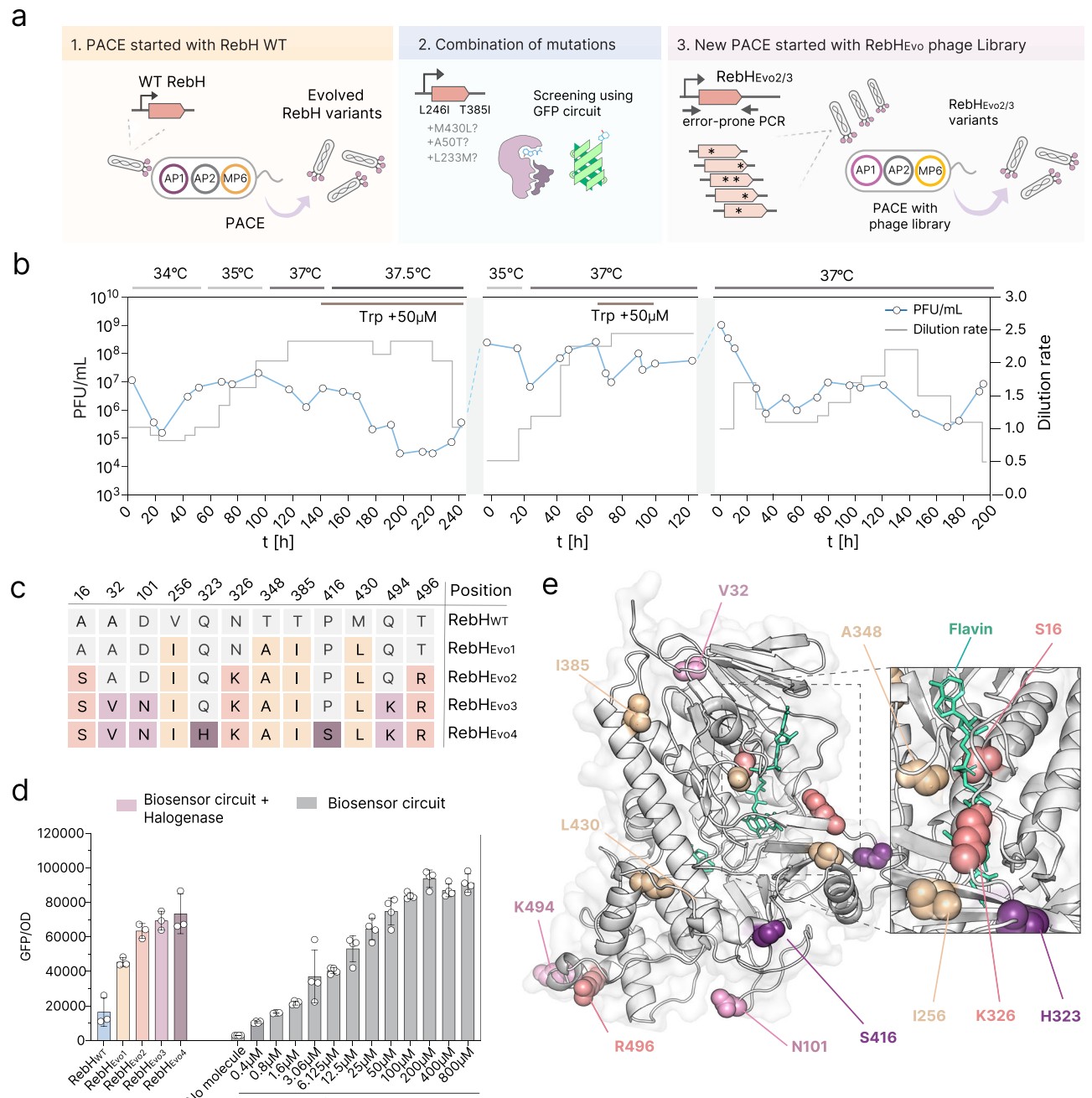

**Fig. 3 | Phage-assisted continuous evolution of RebH halogenase. a** Schematics of the evolution campaign. **b** Phage titres (measured by qPCR), dilution rates, and temperatures during the different PACE runs. PFU/mL stands for plaque forming units per mL. **c** Mutations across RebH during evolution campaigns. **d** GFP/OD$_{600}$ measurement of evolved RebH variants obtained during PACE campaigns, at 37 °C, in DRM:M9 media (1:9). Measurements are displayed side-by-side with direct addition of 7-Cl-Trp to the media, using a strain without RebH (just biosensor) for comparison. Signal measured using DH10B cells. Error bars show mean and SD between 3 biological replicates for RebH variants measurements and 4 biological replicates for the biosensor. **e** Predicted structure of RebH$_{Evo4}$, generated by Alphafold2, with mutations highlighted. Source data for this figure is available in Source data.

fixed mutation (P416S) and a few single-mutant variants were enriched. After a new round of GFP screening and mutation combination (Supplementary Fig. 2E), focusing on the enriched mutants and charged surface residues (such as Q323H), we obtained our final RebH variant, RebH$_{Evo4}$, with a fivefold increase in sfGFP signal compared with the WT enzyme (Fig. 3d, and Supplementary Fig. 2E). However, since our sfGFP circuit response is not linear at high 7-Cl-Trp concentrations (Fig. 3d), we anticipated the real increase in RebH activity to be much higher.

Our final evolved RebH$_{Evo4}$ enzyme contains 12 mutations (A16S, A32V, D101N, V256I, Q323H, N326K, T348A, T385I, P416S, M430L,

Q494K, and T496R), which are spread across the protein structure (Fig. 3e). Some of these mutations were observed at surface residues (Q494K, T496R, P416S, D101N, and Q323H), whereas some face the internal structure of the protein (T385I, A32V, and M430L). Interestingly, at least four mutations were observed around the flavin binding pocket (T348A, A16S, V256I, and N326K), with T348A and A16S having their side chain very close to flavin itself (Fig. 3e).

**Evolved RebH$_{Evo4}$ is more soluble and more active than the WT**
We next sought to characterise our evolved halogenase RebH$_{Evo4}$ in a whole-cell bioconversion assay using HPLC/MS. We observed that

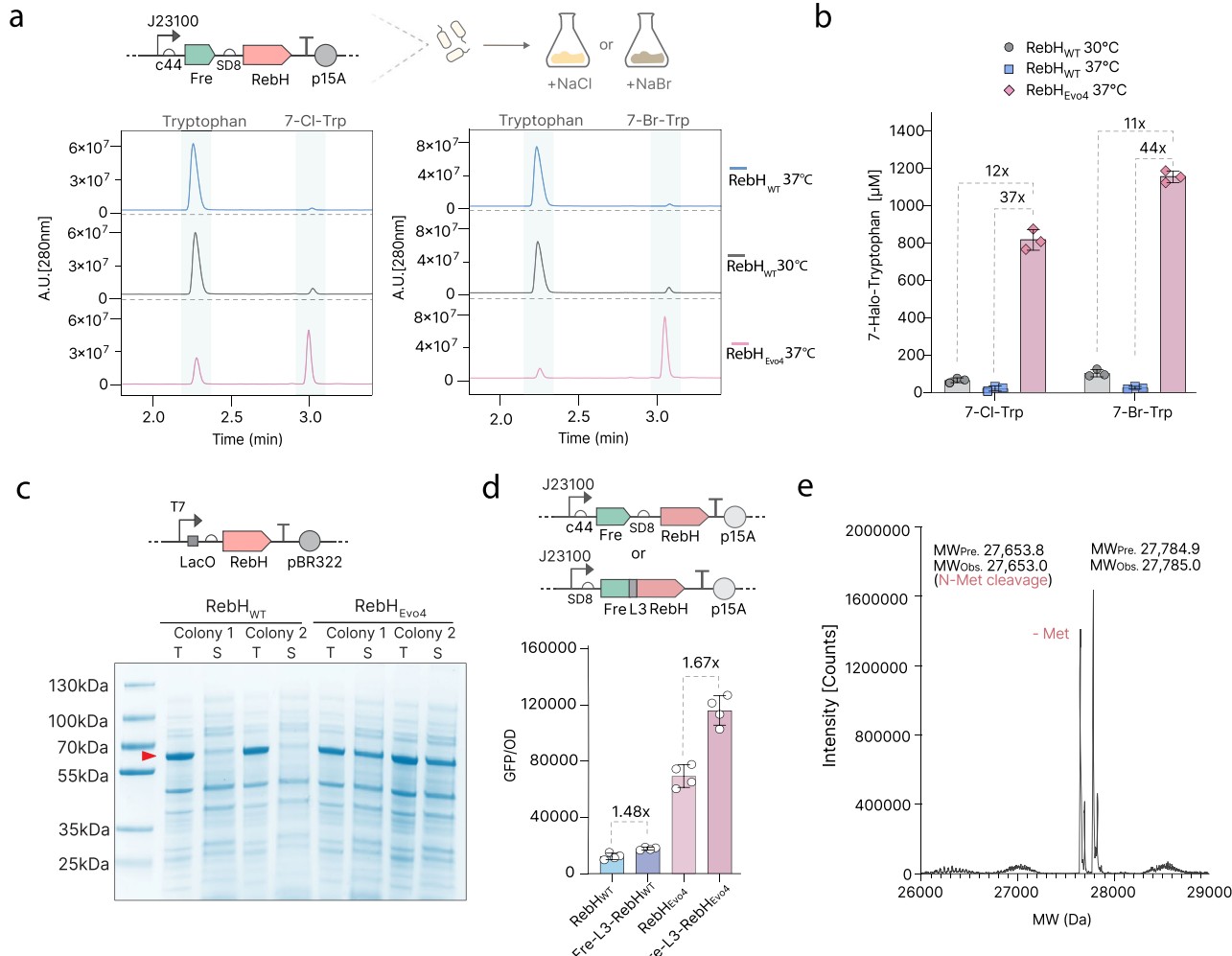

**Fig. 4 | Characterization of evolved halogenase RebH$_{Evo4}$. a** HPLC of whole-cell expressing RebH$_{WT}$ or RebH$_{Evo4}$ at 30 °C or 37 °C, for the production of 7-Cl-Trp (left) or 7-Br-Trp (right). HPLC traces shown are representative of the 3 biological replicates assayed. Raw data can be found in Supplementary Figs. 3 and 4. **b** HPLC quantification of 7-Cl-Trp and 7-Br-Trp production by whole-cell reaction (starting OD$_{600nm}$ of 5). Error bars show mean and SD between 3 biological replicates. **c** Comparison of solubility of RebH$_{WT}$ and RebH$_{Evo4}$ as measured by SDS-PAGE, using BL21 (DE3) strains and T7 promoter. T = Total fraction, S = Soluble fraction.

The image is representative of 2 independent gels prepared on different days. **d** Comparison between Fre-L3 fused and non-fused RebH variants in the GFP biosynthesis (GFP/OD$_{600}$) and biosensor system expressed in the ΔtnaA strain of *E.coli*. Error bars show mean and SD between 4 biological replicates. **e** Whole-protein mass spectrometry of sfGFP produced using Fre-L3-RebH$_{Evo4}$. The correct expected mass for 7-Cl-Trp incorporation at position 151. MW$_{Pre.}$: Predicted molecular weight. MW$_{Obs.}$: Observed molecular weight. Source data for this figure is available in Source data.

strong aeration correlates with good production of halogenated products, so 280 rpm and small volumes per vial were used. ΔtnaA cells encoding Fre and either RebH$_{Evo4}$ or WT RebH were grown to mid-log phase, transferred to a minimal biocatalysis buffer, and allowed to catalyse overnight reactions with saturating tryptophan. RebH$_{Evo4}$ exhibited a 37-fold increase in production of 7-Cl-Trp at 37 °C compared to the WT enzyme (Fig. 4a, b, and Supplementary Fig. 3), demonstrating a substantial gain in overall activity. We also tested the activity of our enzyme at 37 °C versus the WT at 30 °C, which is the optimal temperature for WT RebH expression in *E. coli* due to its poor solubility[7,12]. Reassuringly, the activity of our evolved enzyme at 37 °C still outperformed the WT enzyme at 30 °C by 12-fold (Fig. 4a, b, and Supplementary Fig. 3).

Next, we measured the ability of RebH$_{Evo4}$ to halogenate using bromide instead of chloride. We replaced NaCl with NaBr in the media and found that yields for the brominated product were also increased using RebH$_{Evo4}$, with a 44-fold increase compared to the WT at 37 °C and an 11-fold increase at 30 °C (Fig. 4a, b, and Supplementary Fig. 4).

RebH is known to be an extremely insoluble protein[7]. Since we observed several surface mutations in our evolved mutant, we

investigated whether PACE had affected the overall solubility of the protein. We overexpressed the enzyme variants and observed that RebH$_{Evo4}$ is significantly more soluble than WT as determined via SDS-PAGE (Fig. 4c).

To evaluate whether the activity increase observed for RebH$_{Evo4}$ is solely attributable to improved solubility, or whether its catalysis may also have been enhanced by evolution, we used an established in vitro assay[36] to compare the catalytic rate of purified RebH$_{Evo4}$ with RebH$_{WT}$ at 30 and 37 °C. Indeed, the evolved enzyme showed improved catalysis, with a ~2.5-fold increase in apparent $k_{cat}$ at saturating tryptophan concentrations[30,36] for chlorination at both temperatures, and similar improvements for bromination (Supplementary Fig. 5). The regioselectivity of the evolved enzyme did not change, still exclusively halogenating tryptophan at the C7 position (Supplementary Fig. 6). Finally, we do note that there are other aspects of RebH function we did not test which could have changed as well, such as better dimerisation or reduced substrate inhibition, which are both known to affect halogenase activity[18,37].

The activity of tryptophan halogenases has been previously improved by N-terminal fusion of a flavin reductase[7]. Two mechanisms

may be responsible: improved solubility or creation of a channelling effect between the flavin reductase and the halogenase, thus increasing localized concentrations of $FADH_2$[26], an essential cofactor for RebH. We tested whether N-terminal fusion of a flavin reductase can further improve the activity of RebH$_{Evo4}$, using the previously described L3 linker[7]. Indeed, we found that WT RebH and RebH$_{Evo4}$ are both improved by the fusion, with similar fold-change (Fig. 4d; 1.5-fold for WT and 1.7-fold for RebH$_{Evo4}$), suggesting that such fusion improvements at 37 °C are likely due to solubility-independent channelling effects. We also decided to benchmark our evolved halogenase against two previously described RebH variants evolved for thermostability in vitro[22], using our GFP circuit. Interestingly, neither of the variants tested (3-LR and 3-LSR) seem to be remarkably active when used in a living cell (Supplementary Fig. 7), highlighting the significance of RebH$_{Evo4}$ for efficient in vivo halogenation.

Finally, we tested the efficiency of amber suppression of our evolved enzyme at 37 °C. To improve the signal in this assay, we added a new point mutation, M490L, to the ChPheRS-4$_{*S333C}$ synthetase, which has been reported to increase its efficiency at 37 °C[28]. Running our ΔtnaA strain in a modified M9 medium, we reached 100% amber suppression in cells expressing the full biosensor-biosynthesis circuit with 1 TAG present, and 20% when 3 TAGs were used; the WT enzyme can only achieve 22% amber suppression with 1 TAG (Supplementary Fig. 5A). We then confirmed by mass spectrometry that the sfGFP produced had correctly incorporated 7-Cl-Trp and 7-Br-Trp (Fig. 4e, and Supplementary Fig. 5B). These results indicate that our evolved enzyme enables complete site-specific halogenation at one residue, as well as site-specific halogenation at multiple residues, neither of which was possible with the WT enzyme.

## Efficient production of halogenated biomolecules and peptides using RebH$_{Evo4}$

Enzymatic halogenation is a promising alternative to chemical methods for inexpensive, green production of commodity chemicals. We sought to explore the utility of our evolved RebH$_{Evo4}$ beyond its native product, 7-halotryptophan. Halogenated tryptamines are an important class of biomolecules, which are widely used as drugs for migraines and cluster headaches[38], and have recently begun to attract attention owing to the success of serotonin receptor agonists in combating treatment-resistant depression[39]. Indeed, 7-chlorotryptamine is one of the most potent serotonin 5-HT$_{2A}$ receptor agonists reported to date[40]. Halogenated tryptamines have also been used as precursors for the discovery and production of several active molecules derived from plants, such as alstonine[41] and the anti-cancer precursor strictosidine[42]. Using a 2-plasmid system, we coupled our halogenase and flavin reductase with RgnTDC, a tryptophan decarboxylase, to create a metabolic pathway converting tryptophan into 7-chloro or 7-bromotryptamine[12,43]. Just like with its native product, we observed that RebH$_{Evo4}$ dramatically increased the final yields, with a 24-fold and 36-fold increase in 7-chloro and 7-bromotryptamine production, respectively (Fig. 5a), indicating that the utility of RebH$_{Evo4}$'s improvements is not limited to its native product and may have wider applications in metabolic engineering.

To further explore the titres that our evolved RebH$_{Evo4}$, we scaled up our bioconversion process to a 5L fed-batch bioreactor, feeding a culture in our M9-based minimal medium using relative dissolved oxygen saturation control (rDOS-stat) with a glucose-based feed. Over the course of approximately 30 h at 37 °C, we obtained a final titre of 2.7 g/L (approx. 12 mM) 7-Cl-Trp at (Fig. 5b). To our best knowledge, this represents the highest titre currently reported in the literature[7,12,19,44].

We then decided to apply our evolved halogenase towards the production of halogenated proteins. AMPs have emerged as a promising modality for combating multidrug-resistant pathogens, however they characteristically suffer from instability and low protease

resistance[45]. Tryptophan halogenation is observed widely in natural AMPs and has been demonstrated to improve these traits[14,16]. For example, the AMP Krisynomycin contains two 7-Cl-Trp residues, which are crucial for its potent activity against methicillin-resistant *Staphylococcus aureus*[15,46]. However, engineering halogenation into novel AMPs is challenging since finding positions which tolerate this modification and producing successful variants at scale requires slow-turnaround custom solid-phase synthesis using ncAAs, which is costly, not always suitable for large-scale production, and is not environmentally friendly[47,48]. We reasoned that our evolved RebH$_{Evo4}$ could tackle all of these problems by allowing both screening for active variants and their subsequent large-scale production to be performed autonomously in *E. coli*.

First, we constructed an arabinose-inducible peptide expression system which, in concert with Fre-L3-RebH$_{Evo4}$ and the ChPheRS4 synthetase, allowed us to express peptides containing arbitrary, site-specific 7-Cl-Trp or 7-Br-Trp modifications, thereby enabling a suicide assay for AMP variants that impair growth rate (Fig. 5c). Taking the recently reported AMP Enterocin RJ-11 as a model[49], we rapidly screened for growth-inhibiting activity with chlorination at each of its 3 tryptophan residues (W12, 30, and 38) and observed that modification at position 30 resulted in no significant loss of potency (Fig. 5d).

To confirm these results, we performed a minimum inhibitory concentration (MIC) assay on commercially chemically synthesised versions of WT Enterocin RJ-11 and its Trp30-chlorinated counterpart, and saw no significant difference between their ability to kill *E. coli* cells (Fig. 5e), indicating that the results of our screening approach can accurately predict halogenated AMP activity. Using a previously-reported SUMO-fusion antimicrobial-peptide expression approach[50] (which allows for the expression of AMPs without killing the host cell), we then overexpressed and purified the fusion (Fig. 5f). Following in vitro cleavage and purification of our Trp30-chlorinated variant (Fig. 5g), we verified via High-performance liquid chromatography (HPLC) and mass spectrometry that the product was the correct, chlorinated species (Fig. 5g). Finally, we confirmed that the purified biosynthesised peptide killed cells effectively via the same mode of action (membrane damage) as a commercially chemically-synthesised control (Fig. 5h). Together, these results demonstrate that our evolved halogenase can enable efficient production of active halogenated proteins and peptides, without the need to supplement the media with expensive ncAAs.

## Discussion

Engineered halogenases have long promised the possibility of bio-producing a wide variety of halogenated products, but attempts to overcome the inherent limitations imposed by their poor solubility, temperature stability, and catalysis have been difficult. Using PACE and our GFP biosensor and biosynthesis circuits, we successfully engineered a RebH variant, which to our knowledge, shows the highest activity in vivo of any tryptophan halogenase currently reported, and has improved in vitro catalytic rate compared to the WT. Our final evolved halogenase, RebH$_{Evo4}$, has 12 mutations distributed across its surface, internal structure, and around the flavin binding pocket, which in addition to increasing its activity, make RebH$_{Evo4}$ significantly more soluble than its WT counterpart − to our knowledge, this is also the first reported RebH variant with increased solubility.

Previous efforts to improve halogenases through directed evolution largely employed low-throughput screening of in vitro enzyme activity[22,51,52], where mutations may arise that do not translate well to in vivo use[53]. This is particularly relevant in the case of RebH, which as a metabolically-coupled enzyme, requires multiple other purified enzymes and cofactors to function in vitro, making in vivo use more practical for large-scale, low-cost bioproduction[12,44]. By employing an in vivo, phage-coupled selection using a biosensor, we not only improved the amount of sequence space explored in evolution by

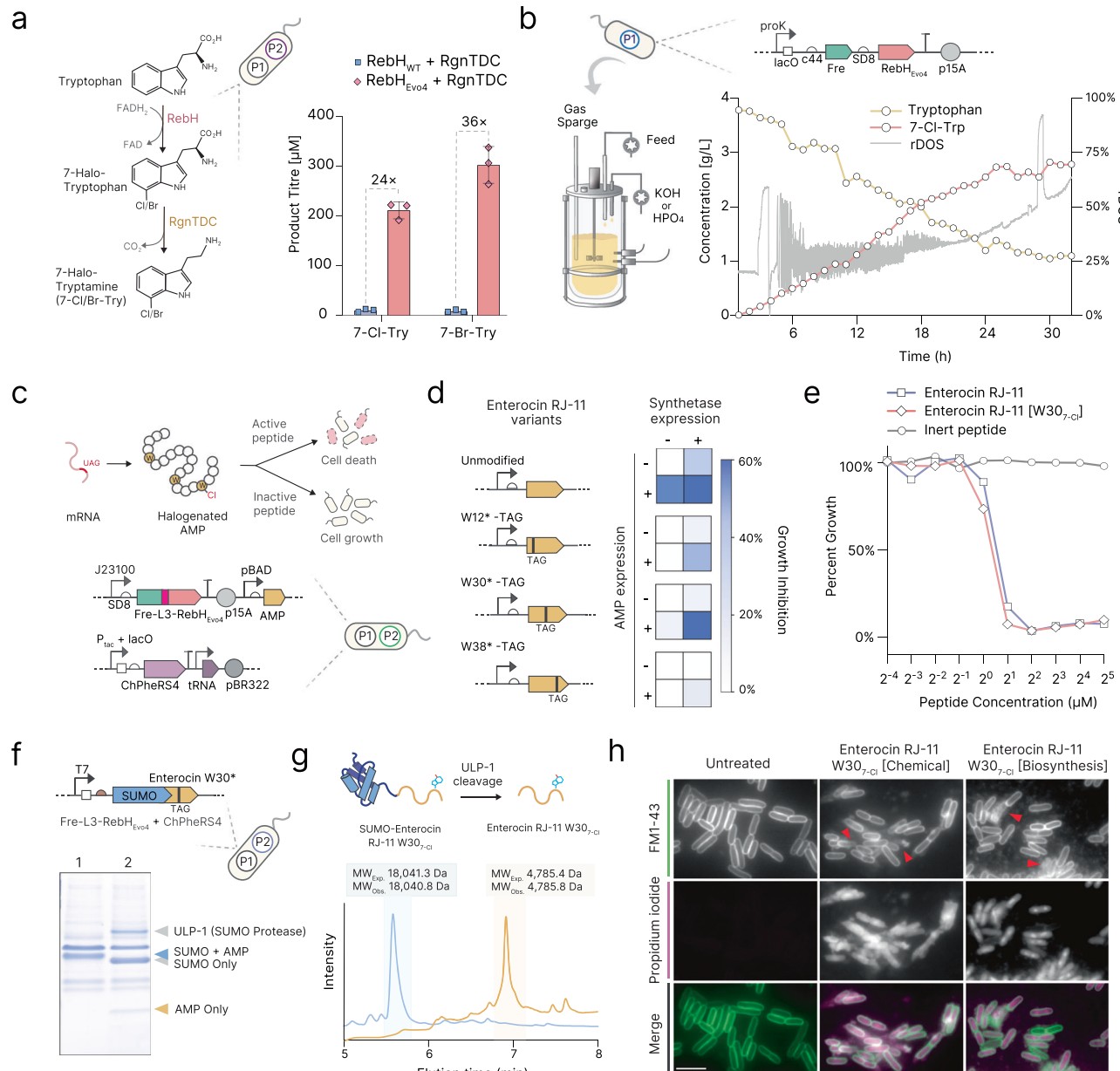

**Fig. 5 | Bioproduction of halogenated biomolecules and halogenated antimicrobial peptides. a** Production of halogenated tryptamine using unfused RebH, Fre and RgnTDC, at 37 °C. Error bars show mean and SD between 3 biological replicates. P1 and P2 represent the plasmids used for the circuit (see Supplementary Table 1). **b** Absolute concentrations (left axis) over time during fermentative production of 7-chlorotryptophan from tryptophan in a fed-batch bioreactor process, at 37 °C. Relative dissolved oxygen saturation (rDOS) used to control feeding throughout the run is also shown (right axis). **c** Schematic of halogenated AMP suicide assay. Halogenated AMP variants are expressed in vivo and active variants identified via growth suppression. P1 and P2 represent the plasmids used for the circuit (see Supplementary Table 1). **d** Growth rate suppression data (% relative to untransformed control strain) for halogenation and truncation at all three Trp residues in the AMP Enterocin RJ-11. **e** MIC assay showing *E. coli* kill curves of commercially-synthesised WT and Trp30-Cl Enterocin RJ-11, alongside an inert

control peptide. **f** SDS-PAGE analysis of halogenated peptide purification intermediates, band identities were determined by comparison with MW markers and commercial peptide standard (for full gel image see Supplementary Fig. 9B). Left lane (1) is whole IMAC eluate, right lane (2) is eluate post SUMO cleavage with ULP-1. P1 and P2 represent the plasmids used for the circuit (see Supplementary Table 1). **g** HPLC/MS of SUMO-peptide pre and post cleavage and purification showing production of the correctly purified halogenated species. $MW_{Pre}$: Predicted molecular weight. $MW_{Obs}$: Observed molecular weight. **h** Fluorescence microscopy assay showing *E. coli* cells treated with commercially chemically-synthesised Trp30-Cl Enterocin RJ-11 and purified biosynthesised product. Cells stained with FM 1-43 and propidium iodide to mark disrupted membranes. Images are representative of cells treated and imaged at least twice, on independent days. Source data for this figure is available in Source data.

multiple orders of magnitude, but also ensured that the mutations we identified would confer real-term activity increases in the intended system of use.

Since halogenation is often the bottlenecking step in biosynthetic pathways[7,13], our demonstration that $RebH_{Evo4}$ is over an order of magnitude (~40×) improved in both halogenated tryptophan and

tryptamine biosynthesis represents an important step towards the sustainable bioproduction of multiple highly relevant halogenated drugs[40], pigments[7], and industrially-useful molecules[8,11–13]. Furthermore, halogenated tryptophan is uniquely also able to be incorporated into peptides and proteins as a ncAA, where it can serve as a stability and activity-enhancing modification. Recent advances in genetic code

expansion enable efficient incorporation of multiple ncAAs into regular and macrocyclic peptides, but still require their direct supplementation to growth media, which is often cost-prohibitive[54–56]. Here, we demonstrate that our evolved RebH$_{Evo4}$ facilitates both in vivo screening and autonomous bioproduction of potent halogenated AMPs without the need for supplementation with expensive ncAAs, paving the way for inexpensive development and biomanufacturing of next-generation peptide antibiotics. While solid-phase peptide synthesis is commonly used, it becomes inefficient for long peptides or proteins and is environmentally unfriendly due to high volumes of solvent and waste[48]. In contrast, biological peptide synthesis has been demonstrated to be cost-competitive, and avoids these environmental concerns. For example, Gaglione et al. demonstrate that in an optimised system, biological production costs can be reduced down to 42 €/mg of final peptide, around 10× less than commercial prices for chemical synthesis strategies[57].

Industrial production of tryptophan itself is routinely performed in modified *E. coli* and *Corynebacterium glutamicum*, reaching titres in the range of 40–50 g/L[58,59], and its halogenated counterparts are often around two to three orders of magnitude more valuable. Our final titre in a 5L bioreactor of 2.7 g/L is therefore encouraging, especially considering that it is likely yields could be further improved, by optimising fermentation conditions and/or using stronger expression to drive halogenation machinery.

The success of our approach may be valuable for the engineering of enzymes more broadly. Most biosensors employed to date for selection and evolution of improved enzyme variants have been transcription factors or riboswitches[60–63], whereas the use of aaRSs as biosensors has remained relatively underexplored. Interestingly, hundreds of different ncAAs have been selectively incorporated into proteins in a highly specific manner using evolved aaRS-tRNA pairs[54,64], which represents a significant untapped opportunity to evolve and engineer a large number of pathways for the biosynthesis of valuable ncAAs or their derivatives. Moreover, unlike transcription factor-based biosensors, aaRS-based readouts enzymatically consume the product they are detecting, potentially increasing their ability to select for highly active enzyme variants by better protecting against diffusion between neighbouring cells. aaRSs are uniquely well-suited to distinguish between similar small molecules, making them a promising avenue for further sensor engineering where precise delineation between multiple possible enzyme products is essential.

We envision our evolved enzyme's improvements in solubility and activity serving as a chassis for next-generation halogenase design. Notably, none of the twelve adaptive mutations in RebH$_{Evo4}$ falls within its tryptophan binding pocket; an interesting future line of work would combine this with previously reported efforts in engineering RebH to alter its substrate scope. For example, previous efforts have shown it is possible to transplant residues between halogenases to alter their regioselectivity[65], raising an intriguing possibility that the improvements of RebH$_{Evo4}$ could be applied to the synthesis of 5- and 6-halotryptophan simply by transplanting known mutations that confer specificity to those positions[52]. One can also imagine combinatorial engineering campaigns marrying the high in vivo activity and solubility of RebH$_{Evo4}$ with machine-learning guided substrate-scope mutations or other aaRS-based biosensors. The remarkable improvements of RebH$_{Evo4}$ remove a key obstacle in the path towards scalable bioproduction of halogenated therapeutics, materials and bioactive peptides.

## Methods

### Bacterial growth conditions, plasmid and phage cloning
Annotated sequence maps of all plasmids and phages used in this study are available as Supplementary Data 1, and selected plasmids are available on Addgene under access numbers 247093 – 247103.

For regular growth, cloning, and bacteria propagation, LB media was routinely used, and the correct antibiotics matching plasmids resistances were added to plates and media. Whenever more than 3 antibiotics were used in combination, the antibiotic concentration was used as 0.5× of the final concentration. For BW25113 ΔtnaA (F$^+$) creation, K-12 BW25113 Δ*tnaA::nptII*(KanR) cells (Keio Collection) were conjugated with S2060 strain (which contains the F-plasmid) and selected against Kanamycin and Tetracycline (marker present in S2060 F-plasmid).

Plasmids were cloned using NEBuilder HiFi assembly (New England Biolabs), and polymerase chain reactions (PCR) were performed using Q5 2X Hotstart mastermix (New England Biolabs). For phage cloning, the phage backbone was amplified using PCR and gene fragments were inserted using NEBuilder. The transformation mix was added to the chemocompetent S238 strain, transformed by heat shock, and then the cells were grown overnight at 37 °C in LB media without any addition of antibiotics. The next day, the culture was spun down and the supernatant filtered through a 0.22 μm syringe filter. The supernatant was then plaqued. Briefly, phage supernatant was serially diluted (tenfold) into 200 μL of DRM media. After that, 50 μL of exponentially-growing (OD$_{600}$ 0.4 to 0.6) S238 cells, which allow activity-independent phage propagation and express lacZ when infected[66], were added, and then a mixture of 550 μL of DRM with 200 μL of melted LB agar, plus Blue-o-Gal (kept at 40 °C using a thermal block) was added to the diluted phage-bacteria mixture. This soft-agar mixture (1 mL) was then quickly pipetted to the top of the wells of a 6-well plate, previously coated with 1 mL of solidified LB agar, and incubated overnight at 37 °C (without inverting the plate) to grow. Individual plaques were then sequenced and grown with S238 strain for propagation, and the sequence-verified phage supernatant kept at 4 °C.

For phage library preparation, a similar method described by Jones and collaborators[67] was used. Briefly, error-prone PCR of RebH was performed using Agilent Mutazyme II kit, and cloned into our phage backbone using NEBuilder. A 100 μL final reaction volume was used for that, aiming to have a 1 μg total of DNA in the reaction. After incubating for 30 min at 50 °C, the assembly reaction was purified using Monarch PCR purification kit (NEB) and eluted in 10 μL of water. For competent cell preparation, 50 mL of S238 cells were grown in 2xYT media until OD$_{600}$ of 0.5. Cultures were then chilled for 10 min in ice and washed twice with ice-cold 10% glycerol, and resuspended into a final volume of 100 μL. Cells were then mixed with the whole purified assembly reaction and transformed via electroporation, and recovered immediately in SOC media. Estimation of the number of cloned phage was performed after 30 to 45 min of recovery. Cells were left to grow overnight, and the next day, supernatant containing the phage library was harvested.

### Amber suppression sfGFP assays
Amber suppression was performed using either BW25113 ΔtnaA strains or DH10B (New England Biolabs) cells. For amber suppression experiments, a mixture of M9 media (M9 + casamino acids (2% w/v final) + glucose (0.5% w/v) and Davis Rich Media (DRM)[68] was used, in a proportion of 9:1. DRM is the optimised media generally used for PACE, however we observed that in terms of raw circuit performance, M9 gave higher signal on GFP readout. To ensure high circuit performance without compromising phage activity, a 1:9 and 5:95 ratio of the two media was tested initially, and both already gave good fold-change in signal, with good growth. Since we obtained robust growth at 10:90, we decided to move on with this ratio. Cells were pre-grown into M9/DRM mixture to an exponential phage and then inoculated into fresh media, supplemented with 0.5 mM IPTG (final concentration) and the corresponding antibiotics. Cells were then grown overnight (at different temperatures, as determined by the experimental conditions), for 14–16 h, spun down and washed twice (8000 × *g*, 3 min) with PBS, before reading the sfGFP signal using a TECAN plate reader, using a 96-well plate. Raw GFP signals were then normalised by the OD$_{600}$

readings to ensure that changes in growth rate do not impact the fidelity of measurements. We observed that it is important to keep the ON incubation for around 14–16 h, to avoid oversaturation of GFP signal and alterations in GFP/OD ratio between different mutants.

## Phage propagation assays and quantification

Phage propagation was estimated either by plaquing or by using qPCR to estimate phage enrichment. To estimate 7-Cl-Trp-dependent phage propagation, BW25113 ΔtnaA (F⁺) cells were used, with overnight incubation. After incubation, phage suspension was collected, filtered and plaqued. For plaquing, the same method was used as described for phage clonings, always as an overnight incubation at 37 °C. For qPCR phage estimation, 0.75 μL of phage suspension was combined with 5 μL of PowerUp™ SYBR™ Green Master Mix (Thermo Fisher Scientific), 0.0625 μl each of 100 μM M13 forward and reverse primers (5'-CACCGTTCATCTGTCCTCTTT-3' and 5'-CGACCTGCTCCATGTTAC TTAG-3'), and water to achieve a final volume of 10 μL. The qPCR was performed with the following conditions: 95 °C for 2 min, and then 40 cycles of 95 °C for 15 s and 60 °C for 60 s. To generate a standard curve for qPCR, a standard phage sample of high titre (-1 × 10¹⁰ PFU/mL as determined by plaquing) was serially diluted by a factor of 10, up to 10⁷-fold, in water. A standard curve was generated using Cq values, and phage titres were determined accordingly.

## Phage-assisted continuous evolution (PACE)

PACE was performed using similar methods as described previously[66], with a few alterations. In our PACE system, instead of using a chemostat providing new host cells to the lagoon, we used a cell culture kept at 4 °C, in a sterile flask inside a fridge. BW25113 ΔtnaA (F⁺) cells (BW25113 ΔtnaA containing the F plasmid) were freshly transformed with the MP6 mutagenesis plasmid[33] and grown in M9:DRM media (9:1 ratio) until reaching $OD_{600}$ 0.05 to 0.1, then quickly chilled on ice and placed at 4 °C. The cell culture flask is connected to the lagoon (placed inside an incubator) by tubings attached to peristaltic pumps, similarly as traditional PACE set-ups. Magnetic stirrers were placed inside both the 37 °C lagoon and the 4 °C culture flask to keep both evenly suspended. We observed that bacterial cells would grow and divide inside the tubing, especially in the section outside of the fridge and in the section inside the 37 °C incubator, before reaching the lagoon, giving us a higher final $OD_{600}$ (around 0.25 to 0.5) inside the lagoon. Culture media (same as used for cell growth) with inducer (L-arabinose 0.4% w/v) were kept in a separate flask at room temperature and pumped into the lagoon in parallel to fresh bacterial culture, to delay induction of the mutagenic proteins encoded on MP6 until cells reached the lagoon (final L-arabinose concentration of 0.2% w/v). The initial phage suspension for each PACE run (10⁷–10⁹ PFU/mL) was inoculated directly into the lagoon. For the first PACE run, a clonal phage population of phage containing WT RebH was used. For the second and third PACE runs, phage libraries were used (Fig. 3). Samples were collected from the lagoon in intervals using a sterile needle, and the phage population was monitored during the runs using qPCR (as described in 'Phage propagation assays and quantification'). Mutations were evaluated by amplifying and sequencing the RebH gene from the phage, either from plaques or from the entire lagoon phage population.

## RebH solubility assays

To test RebH solubility, we cloned the WT sequence and our evolved variant into a pET28a vector (pAP108 and pAP109, Supplementary Table 1). The His-Tag sequence was removed. We then transformed E. coli BL21(DE3) cells, and grew independent colonies in 2xYT until reaching $OD_{600}$ 0.4. Protein expression was then induced for 3 h by adding IPTG (1 mM) to the media. After induction, cell pellets were harvested for protein extraction. Total protein fraction and soluble fraction were prepared by using the exact same volume of culture[7]. For the total protein fraction, cells were pelleted and resuspended in a 1×

Laemmli buffer (BioRad). For soluble protein fraction, cells were pelleted, frozen and thawed and resuspended in B-PER protein extraction reagent (Thermo Scientific). Extraction with B-PER (Thermo Scientific) was performed at RT for 1 h at a rotary well. The extract was then centrifuged (16,000 × g for 5 min), and the supernatant was collected. Samples were loaded onto an SDS-PAGE gel (4–18%), run using MOPS buffer and stained using Coomassie blue.

## HPLC/MS quantification of 7-halotryptophan and 7-halotryptamine bioproduction

For HPLC/MS estimation and quantification of 7-halotryptophan and 7-halotryptamine production, E. coli cells (BW25113 ΔtnaA) were freshly transformed with the respective plasmids containing the WT RebH and RebH_Evo4 sequences. Both systems used the same genetic arrangement in an operon, with J23100 as a promoter and RBSc44 for Fre and SD8 for RebH variants. (See plasmid pAP65 and pAP113, Supplementary Tables 1 and 2). For 7-halotryptamine assays plasmid pJB21x05 encoding RgnTDC was additionally co-transformed. Colonies from each plate were collected and inoculated into LB media either at 37 °C or at 30 °C and incubated at 230 RPM. Cells were then grown to an $OD_{600}$ of 0.55 (±0.05), pelleted and resuspended to an $OD_{600}$ of 5 in the final M9-based whole cell catalysis buffer. Base Cl-free M9-derivative (Buffer v4.1): $Na_2HPO_4$ (47.8 mM), $KH_2PO_4$ (22.0 mM), $(NH_4)_2SO_4$ (9.4 mM), casamino acids (0.2% w/v), glycerol (0.4% v/v), glucose (25 mM). Additionally, 5 mM tryptophan substrate was used in all assays, and 200 mM NaCl or NaBr was used depending on the intended product. We observed that strong aeration correlates with good production of halogenated products. The final volume of the reaction was 2 mL. Cells were incubated in 50 mL falcon tubes, overnight at 280 RPM either at 37 °C or 30 °C.

Samples were centrifuged for 10 min at 4000 × g and 30 μL of the supernatant were taken for UPLC/MS analysis. LC-MS data were obtained on a Waters ACQUITY (Massachusetts, USA) equipped with QSM, QDa and PDA detectors, sample manager FTN-H, quaternary solvent manager, column manager with ACQUITY UPLC BEH C18 1.7 μm, 2.1 × 50 mm column. Electrospray ionisation (ES+ and ES-) and Diode Array spectra were obtained for each characterised compound. The gradient methods for LC-MS were composed of 0.1% formic acid (FA) in water/0.1% FA in acetonitrile, over 8 min. Peak identity was verified by Mass Spectrometry, and quantification of the peaks (280 nm) was performed by comparing peak area to a standard curve obtained by running commercial standards of 7-Cl-Trp and 7-Br-Trp. Experiments were performed in triplicates.

## sfGFP purification and MS analysis

For sfGFP purification and MS, E. coli cells (BW25113 ΔtnaA) transformed with a plasmid containing RebH_Evo4 and ChPheRS-4*S333C + M490L were grown into LB media until $OD_{600}$ 0.1. Cells were spun down, washed 2× in PBS and resuspended in a 20 mM sodium phosphate buffer, with either 200 mM NaCl or 200 mM NaBr, 1% glucose and 2 g/L of casamino acids. IPTG was added to a final concentration of 0.5 mM. Cells were grown overnight at 37 °C. The next day, the pellet was collected, and GFP was extracted using B-PER (Thermo Scientific) and purified using HisPur Ni-NTA resin (Thermo Scientific). The purity of the purified sfGFP was analysed using sodium dodecyl sulfate–polyacrylamide (SDS-PAGE) gels, followed by staining with Coomassie Blue. The sfGFP was then analysed by MS, and the intact mass analysed.

## Protein production and purification of RebF

In vitro assays of RebH require a flavin reductase, such as RebF (UniProt Q8KI76), for in situ regeneration of the $FADH_2$ cofactor[30,36]. For RebF production and purification, we adapted the procedure established by Payne et al.[36] using a MBP-fusion to improve RebF solubility and protein yield, as described in the following. Chemically competent

BL21(DE3) cells (New England Biolabs) were transformed with pASx48a (Supplementary Table 1) and precultured in LB with 50 µg/mL kanamycin before transfer to a 1 L culture in buffered Terrific Broth (12 g/L tryptone, 24 g/L yeast extract, 0.4% v/v glycerol, 0.017 M KH$_2$PO$_4$, 0.072 M K$_2$HPO$_4$) supplemented with 50 µg/mL kanamycin. The culture was shaken vigorously at 37 °C in a 5 L shake flask and grown to mid-log phase. Upon reaching an OD$_{600}$ of 0.6, the culture was briefly cooled on ice and IPTG (isopropyl β-D-thiogalactopyranoside) was added to a final concentration of 0.5 mM to induce RebF production. Growth continued for 12 h at a lowered temperature of 30 °C with vigorous shaking.

Cultures were cooled on ice and harvested at 8 °C, by centrifugation at 8000 × g for 10 min. For lysis, cells were resuspended in cold buffer (25 mM HEPES (pH 7.4), 100 mM NaCl, 10 mM imidazole, 0.01 mg/mL DNase, 6 mM MgCl$_2$, 0.01 mg/mL RNase and 0.6 mg/mL lysozyme) on ice using 3 mL buffer per gram cells and passed three times through a pre-cooled (4 °C) French Press at 18,000 psi. The cell lysate was clarified by centrifugation at 30,000 × g for 30 min at 4 °C.

Affinity purification was performed with HisPur™ Ni-NTA Resin (Thermo Scientific), as follows. Clarified lysate was mixed with resin and incubated for 1 h at 4 °C with mixing. Then, resin was transferred to a gravity-flow column and washed with wash buffer (HEPES 25 mM (pH 7.4), 100 mM NaCl, 25 mM imidazole). Protein was eluted in 25 mM HEPES (pH 7.4), 100 mM NaCl, with 250 mM imidazole. The eluted protein was concentrated using a centrifugal filter with a 30 kDa cut-off (Vivaspin, Sartorius), and the buffer was exchanged five times with Storage Buffer (HEPES (25 mM, pH 7.4), glycerol 10% v/v). NaCl was omitted to remove Cl⁻ ions which may interfere as an undesired halide in downstream bromination assays. Protein concentrations were measured using the Pierce® BCA Protein Assay Kit, and protein aliquots were stored at −70 °C until use. Protein purity was assessed via SDS-PAGE.

## MBP-RebF in vitro activity assay

To verify the stand-alone activity of the coupling enzyme RebF (MBP-RebF fusion), we photometrically measured the rate at which it oxidizes NADH in the presence of FAD. Following a previously established assay[36], the NADH concentration was monitored based on decreasing absorbance at 340 nm. Reactions were performed in a 96-well microtiter plate containing 1 µM MBP-RebF with 1 mM NADH, 50 µM FAD, and 10 mM NaCl in HEPES (25 mM, pH 7.5). Absorbance was monitored at 340 nm using a Tecan Spark plate reader. The observed mean activity of 56 min⁻¹ was similar to prior reports[36] and was sufficient for use as a coupling enzyme for FADH$_2$ co-factor regeneration in RebH assays.

## Protein production and purification of RebH variants

RebH$_{WT}$ and RebH$_{Evo4}$ genes were amplified by PCR from pAP108 and pAP109, respectively, and subcloned into a pET-28a vector by NEBuilder® Assembly, resulting in an N-terminal 6xHis-tag (MGHHHHHHTS). Payne et al.[36] established that purified protein yields of recombinant RebH produced in *E. coli* are significantly improved by co-expression of the chaperones GroES and GroEL (from the pGro7 plasmid). We followed this strategy but chose to use the cold-adapted, constitutively expressed chaperonins provided in the 'ArcticExpress' system (Cpn10 and Cpn60 from Oleispira antarctica), as an alternative to the pGro7 plasmid utilised by Payne and collaborators[36].

ArcticExpress(DE3) (Agilent) were transformed with pASx49 (pET-28a_RebH$_{WT}$) or pASx50 (pET-28a_RebH$_{Evo4}$), respectively. Transformants were selected on LB plates with 50 µg/mL kanamycin and 20 µg/mL gentamicin. The resulting strains were precultured in LB with 50 µg/mL kanamycin and 20 µg/mL gentamicin before transfer to a 1 L culture in buffered Terrific Broth (12 g/L tryptone, 24 g/L yeast extract, 0.4% v/v glycerol, 0.017 M KH$_2$PO$_4$, 0.072 M K$_2$HPO$_4$) supplemented with 50 µg/mL kanamycin and 20 µg/mL gentamicin. The culture was shaken vigorously at 37 °C in a 5 L shake flask and grown to mid-log phase. Upon reaching an OD$_{600}$ of 0.6, the culture was briefly cooled on ice and IPTG (isopropyl β-D-thiogalactopyranoside) was added to a final concentration of 0.5 mM to induce RebH production. Growth was continued for 16–24 h at a lowered temperature of 20 °C with vigorous shaking.

Cell harvest, lysis and purification of RebH were performed similarly to RebF (described in 'Protein production and purification of RebF'), with the following changes: After standard washing of the Ni-NTA resin, an additional wash step for specific dissociation of chaperones[69] (Cpn10 and Cpn60) was performed by incubating the resin in chaperone wash buffer (HEPES 25 mM (pH 7.4), 100 mM KCl, 25 mM imidazole, 10 mM MgCl$_2$, 5 mM ATP) for 1 h at 4 °C with mixing. Efficient removal of residual/co-purifying chaperones was later confirmed by SDS-PAGE. We note that we were able to resolve Cpn60 and RebH protein bands by accurate SDS-PAGE despite very similar molecular size (Supplementary Fig. 6B). Protein was eluted (HEPES 25 mM (pH 7.4), 100 mM NaCl, 250 mM imidazole) and subsequently desalted for 12 h at 4 °C in dialysis tubing with a 10 kDa molecular weight cut-off using 1 L of salt-free storage buffer (HEPES 25 mM (pH 7.4), glycerol 10% v/v). Dialysis was chosen for both RebH variants as a mild desalting method due to observed protein precipitation when rapid desalting or strong sample concentration were performed via centrifugal filters. Protein purity was assessed using SDS-PAGE.

## RebH in vitro activity assay

To determine RebH activity, we adapted a previously established assay[36] to measure substrate conversion by RebH under saturating substrate concentrations. Reactions were initiated containing 25 mM HEPES (pH 7.4), 150 µM L-tryptophan (>20× above the reported $K_m$[36]), 0.5 µM RebH enzyme and halide (10 mM NaCl or 100 mM NaBr, respectively). Co-factor regeneration was achieved by the addition of NAD (100 µM), FAD (5 µM), MBP-RebF (2.5 µM, i.e. 2.5× excess over RebH), glucose dehydrogenase (50 µ/mL), and glucose (20 mM). Glucose dehydrogenase was purchased from Sigma Aldrich (catalogue ID 19359). Reactions were performed in triplicate in 100 µL total volume at 30 °C or 37 °C (as indicated) using a 96-well microtiter plate with orbital shaking at 300 rpm. Time-points were collected at 2–30 min and quenched by addition of an equal volume of MeOH (100%). Precipitation was removed by centrifugation. Reactants and products were analysed by HPLC (as described in 'HPLC analysis of in vitro enzyme assay products') and compounds were quantified against standard calibration curves. Initial reaction rates were measured and used to estimate the apparent $k_{cat}$. We report activities as apparent $k_{cat}$ based on the fact that all substrates were provided at saturating concentrations as determined in past work[30,36].

## HPLC analysis of in vitro enzyme assay products

Products were analysed using high-performance liquid chromatography (HPLC) on a LC-2060C 3D UHPLC (Shimadzu) using a Poroshell 120 EC-C18 4.6 × 100 mm, 2.7 µm pore column (Agilent InfinityLab) with a Poroshell 120 EC-C18 4.6 × 5 mm, 2.7 µm pore guard column (Agilent InfinityLab). The following gradient was used with a flow rate of 1.2 mL/min (mobile phase A = H$_2$O; mobile phase B = MeOH): 95% A/5% B at 0 min, linear gradient to 45% A/55% B at 5.5 min, linear gradient to 100% B at 6 min. In-between samples, the column was flushed with 5 column volumes of mobile phase B and re-equilibrated with 5 column volumes of 95% A/5% B.

10 µL samples were injected and data recorded on a PDA detector at wavelengths 200–500 nm. The column oven and PDA detector were set to 40 °C while samples were kept at 15 °C. Data was analysed in Shimadzu LabSolutions 5.132. For signal processing, chromatograms at 280 nm UV were produced with a 4 nm bandwidth (±) from the PDA data, and the Chromatopac algorithm was used with the 280 nm UV chromatograms for peak integration.

Standards of 7-chloro-L-tryptophan, 7-bromo-L-tryptophan and L-tryptophan were used to identify retention times and to obtain calibration curves for the determination of absolute concentrations. An FAD standard as well as a reaction matrix (lacking RebH) was also analysed, and no interference was found at the relevant retention times or absorbance wavelengths for any of the analysed compounds.

## Fed-batch bioreactor production of 7-Cl-Trp

Scale-up to 5L bioreactor cultivation closely mirrored initial overnight shake-flask development, with a few modifications. The IPTG-inducible high-expression promoter proK-lacO was cloned upstream of the Fre, RebH operon to mitigate toxicity associated with constitutive expression. To initiate the bioreactor runs a pre-culture of freshly-transformed cells was inoculated in 24L in a 100L Eppendorf Bioflo 610 bioreactor, with appropriate antibiotics and 25 mM glucose in Terrific Broth until $OD_{600}$ 0.50, whereupon cells were induced with 0.5 mM IPTG and allowed to grow again until they reached $OD_{600}$ 1.00. 24 L of the resulting pre-culture was spun down at $7340 \times g$ for 30 min at 25 °C (2L Sorvall RC 12BP+ centrifuge) and resuspended to an $OD_{600}$ of 4.8 in 5L of our M9-based biotransformation buffer v4.1 (described in 'HPLC/MS quantification of 7-halotryptophan and 7-halotryptamine bioproduction') and transferred to the final bioreactor (10L Sartorius Biostat C bioreactor). Additional components used for 7-Cl-Trp fermentation: tryptophan (20 mM), IPTG (0.5 mM), NaCl (200 mM). 1 mL samples were taken manually every hour, and $OD_{600}$ recorded. Feed consisted of glucose (500 g/L), $(NH_4)_2SO_4$ (100 g/L), and $K_2HPO_4$ (5 g/L), and dosed via automatic feedback control via rDOS probe, with a programmed setpoint of 25%. Samples were taken manually every hour. Since we had previously noticed that we had been reaching the limits of 7-Cl-Trp solubility at the titres observed by the end of the run, pellets were separated, extracted using acetonitrile, and unified with supernatant to account for all product that had precipitated prior to HPLC/MS quantification.

## Growth-based screening of halogenated antimicrobial peptide variants

Plasmids containing halogenation machinery plus the CDS of AMP variants with Trp residues selected for halogenation replaced with TAGs were cloned, with peptide CDSs under the araBAD promoter and bearing an SD8 RBS (see pJB15x03-series in Supplementary Table 1). Colonies of NEB 10-beta cells freshly cotransformed with these plasmids plus the synthetase machinery plasmid (see pAP80 in Supplementary Table 1) were grown until $OD_{600}$ 0.50 in M9 + Casamino acids (0.2% w/v) and Glycerol (0.4% v/v), supplemented with 25 mM Glucose and appropriate antibiotics, then diluted 100X into 150 μL of the same base medium ±0.5 mM IPTG and 10 mM Arabinose in 96-well optical flat bottom reader microplates (Corning). Growth curves were measured over 10 h of continuous shaking at 37 °C on a Tecan Spark microplate reader.

## Autonomous expression and purification of halogenated Enterocin RJ-11

Plasmids bearing halogenation machinery plus the CDS for Trp30-chlorinated Enterocin RJ-11 fused C-terminally to a 6×His-SUMO tag were cloned, with the SUMO-peptide CDS under the T7 promoter and bearing an SD8 RBS (see pJB18x02v1 in Supplementary Table 1). C321.ΔA T7RNAP ompT- rne- lon- cells (Addgene #182778[70]) were cotransformed with this plasmid plus the synthetase machinery plasmid (see pAP80 in Supplementary Table 1), and were grown until $OD_{600}$ 0.50 in M9 + Casamino acids (0.2% w/v) and Glycerol (0.4% v/v), supplemented with 25 mM Glucose and appropriate antibiotics, at 30 °C. SUMO-peptide expression was induced overnight with 0.5 mM IPTG, and the pellet was harvested, resuspended using B-PER Complete Lysis Buffer (Thermo), and sonicated. SUMO-peptide was purified from clarified lysate using Ni-NTA magnetic beads (NEB), cleaved with ULP-1 (Sigma) for 1 h at 30 °C, and then reverse-purified using a second round of IMAC. The final purified peptide was buffer-exchanged into deionised water using a 3MWCO centrifugal filter (Sartorius). The samples were analysed by HPLC/MS on an Agilent 1260 LC-MSD (Agilent Poroshell 120 EC-C18 2.7 μm, 3.0 × 30 mm column), using a linear gradient of 5–100% B over 8.5 min at a flow rate of 0.425 mL/min. A binary solvent system [A: and H2O /0.08% TFA/1% MeCN and B: MeCN /0.08% TFA] was used.

## MIC assays for antimicrobial peptides

*E. coli* cells (NEB 10-beta) were grown to $OD_{600}$ 0.50 in LB without antibiotics, then diluted 100X into 300 μL LB in round-bottom 96-well deep culture plates, supplemented with a dilution series of the appropriate peptide. Commercially-synthesised peptides (Biosynth) were prepared at 128 μM in deionised water. Plates were incubated overnight at 37 °C with 280 RPM shaking covered with a Breathe Easier (Sigma) membrane, then 100 μL of each was transferred to a 96-well optical flat-bottom reader microplate (Corning) and $OD_{600}$ was measured on a Tecan Spark microplate reader. Measurements were normalised as a percent relative to growth and sterility controls included in every assay.

## Fluorescence microscopy membrane integrity assays

For fluorescence microscopy observations, *E. coli* cells (K-12 BW25113) were grown in LB media until the exponential phase (OD 0.5). Cells were then treated with either the commercial Enterocin RJ-11_{W307-Cl} or the biosynthetized and purified peptide at 2× MIC value (~3 μM) and incubated for 15 min at 37 °C with shaking. Cells were then harvested (1 mL), centrifuged for 3 min at $4000 \times g$ and resuspended in 100 μl of residual media. Cells were treated with 50 μg.ml$^{-1}$ of FM1-43 (Invitrogen™) and 10 μg.ml$^{-1}$ of propidium iodide for membrane staining and permeability evaluation, respectively. About 5 μl to 10 μl of cell suspension were added on top of LB 25% agarose pads, covered with coverslips and imaged using a Nikon Ti2 Eclipse inverted microscope, with a 100× oil objective. Images were analysed using ImageJ.

## Statistics and reproducibility

All images of SDS-PAGE gel analysis presented are representative of experiments performed at least twice, on separate days. Unless otherwise specified, all data points presented individually represent biological replicates picked from separate bacterial colonies. Where error bars are shown, these represent the mean and SD. No datapoints collected were excluded in their calculation. Fluorescence assays for membrane integrity were repeated on cells grown from separate bacterial colonies and on multiple days, with similar results. All images shown are representative. The uncropped images are provided in Source Data.

## Reporting summary

Further information on research design is available in the Nature Portfolio Reporting Summary linked to this article.

# Data availability

Source data are provided with this manuscript. Annotated sequence maps of all plasmids and phage used in this study are available as Supplementary Data 1, and selected plasmids are available on Addgene under access numbers 247093 – 247103. Sequences of evolved proteins are available at NCBI accession PZ059835, PZ059836, PZ059837, PZ059838. Source data are provided with this paper.

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

## Acknowledgements

This work was supported by the Francis Crick Institute which receives its core funding to E.D. from Cancer Research UK (CC2239), the UK Medical Research Council (CC2239), and the Wellcome Trust (CC2239). The authors gratefully acknowledge Svend Kjaer and Vangelis Christodoulou at the Francis Crick Institute Structural Biology STP, as well as Brian Lally and Celine Bouchoux for assistance and advice on protein production and purification. We also thank Proteomics STP, Chemical biology STP and Fermentation STP at the Francis Crick Institute for their support with this manuscript. Field Foundry Ltd (London, UK) performed HPLC method development and quantitative HPLC measurements.

## Author contributions

A.A.P. and E.D. conceived the study. A.A.P., J.B., A.S., and E.D. designed the experiments. A.A.P., J.B., A.S., and L.N. cloned the plasmids. A.A.P. created and calibrated the microbial circuits, phage circuits, performed phage cloning and evolution experiments. O.A. and S.A. established phage quantification methods and PACE set-up. A.A.P., J.B., A.S., and C.S. performed HPLC/MS measurements. J.B. and J.C. performed protein/peptide purification and MS. A.S. performed protein purification, designed and analyzed in vitro assays. J.B., N.P. and A.A. performed bioreactor experiments. J.B. performed antimicrobial peptide screening and purification. A.A.P. and J.B. performed microscopy. Figures were prepared by A.A.P. and J.B. with input from all other authors. Text and were prepared by A.A.P., J.B. and E.D. with input from all other authors.

## Funding

## Competing interests

The authors declare no competing interests.
