## [Peer Review File · Nature Communications]

Continuous evolution of a halogenase enzyme with improved solubility and activity for sustainable bioproduction

Corresponding Author: Dr Erika DeBenedictis

Version 0:

Reviewer comments:

Reviewer #1

(Remarks to the Author)

The authors describe in their manuscript a strategy for the continuous evolution of a halogenase, based on the tryptophan-7-halogenase RebH. RebH is a flavin-dependent tryptophan halogenase for which it has been shown that the introduction of a chloro- or bromo-substituent at position 7 of the indole ring of tryptophan is feasible, and that this biocatalysis can also be performed on a gram scale (*Angew. Chem. Int. Ed.* 2015, 54, 298). Unfortunately, the authors did not cite this work.

Instead, they state that halogenase biocatalysis has shown only limited success because no substantial amounts of halogenated tryptophans can be produced. This overlooks strategies to obtain halogenated tryptophans either in vitro through cell-free systems (*vide supra*) or in vivo by metabolic engineering (see Ref. 39).

As a real highlight, the authors report the use of a sensor system based on amber suppression to detect the formation of halogenated tryptophan in vivo. This system relies on a tRNA synthetase that selectively incorporates 7-chlorotryptophan into proteins at TAG codons. The strategy of genetic code extension was combined with the biosynthesis of chlorotryptophan-containing GFP, where fluorescence is observed only upon incorporation of 7-chlorotryptophan. It was also noted that non-halogenated tryptophan acts as an inhibitor of this synthetase.

This GFP readout was then coupled with a continuous phage-assisted evolution (PACE) approach. After 240 hours under varying temperature conditions, thermostable variants of the halogenase were obtained, albeit with only marginal thermostability. Monitoring individual clones over the course of the selection provided further insight into possible substitutions; in addition, error-prone PCR was applied to the selected mutants, and further clonal variants were isolated in a third PACE phase.

This yielded the variant RebH-Evo4, which is several-fold more active than the wild type. It carries 12 mutations, some located at the protein surface, others in the core. At least four mutations are positioned near the flavin-binding site, but none close to the tryptophan-binding site. In this context, it remains an open question whether the regiochemistry of tryptophan halogenation at C7 of the indole ring is affected, even though the mutated residues are not located directly in the tryptophan-binding region. Unfortunately, the work of the Niemann group (*J. Biol. Chem.* 2019, 294, 2529) on the influence of indole positioning in front of the tunnel between the flavin-binding site and the tryptophan pocket was not cited.

In summary, the authors convincingly demonstrate that a highly active and more soluble halogenase variant could be obtained within a short time by using PACE. They also examined bromination by replacing sodium chloride with sodium bromide, confirming activity, although no data on halide selectivity are presented.

The next section of the manuscript, describing the incorporation of halogenated tryptophan derivatives via genetic code expansion, is a logical continuation of the topic, since the necessary enzymes and methods had already been well established. However, the novelty of this approach is not fully convincing, as the incorporation of non-canonical tryptophan derivatives, including halogenated tryptophans, has already been reported in the literature (see, for example, Kwon & Tirrell, *J. Am. Chem. Soc.* 129, 2007, 10431, and Budisa et al., *Biol. Chem.* 385, 2004, 191). It has nevertheless to be acknowledged that the selection system for antimicrobial peptides is well chosen.

Overall, this is a highly interesting manuscript in the field of enzymatic halogenation, with potential significance for

biocatalysis and sustainable bioproduction. The optimization and selection system is well designed, innovative, and highly efficient. The application to ribosomally synthesized antimicrobial peptides is also convincing, even though the incorporation of halogenated tryptophans in proteins using genetic code expansion has been described previously. However, the manuscript suffers somewhat from incomplete citation.

Questions also remain open: for instance, whether surface mutations affect the oligomerization state of RebH, or whether the mutations influence the halogenation regiochemistry and halide selectivity.

In Figure 4, panel a, a slight shift in retention time between products of RebH wild type and RebH-Evo4 is noticeable, which may indicate altered regiochemistry. The problem of product inhibition in enzymatic halogenation by RebH is also not addressed in the discussion.

In Figure 5g, determination of the exact mass of the respective peptides would have been desirable. Finally, the shoulder in the HPLC analysis (Supplementary Figure 6g) of the halogenated enterocin RJ11 derivative also requires explanation.

The presented work utilizes a sophisticated evolution strategy to achieve impressive increases in productivity and solubility of the halogenase RebH. Overall, the results are very well presented and supported with the data shown. Apart from two comments regarding additional data, I only have minor comments:

1. An in vitro comparison of the new variant RebHEvo4 to the respective wild type would help to further validate the claimed improvement in activity and stability. Based on the in vivo assay, it is not entirely clear if the increase in product formation can be solely attributed to a better solubility in the cellular environment or if an increased activity of the enzyme contributes to this. Although you make a compelling case for the advantages of using halogenases in vivo, it would be interesting to see if the observed advantages of this new mutant translate to smaller scale in vitro use.
2. Even if the binding site is not affected, a total of 12 mutations could potentially affect the global fold and therefore even substrate binding. Could incorporation of e.g. C5- or C6-halotryptophans by ChPheRS-4 be ruled out entirely in this case? If not, NMR-analysis of the halogenated products would be beneficial to confirm the regioselectivity of RebHEvo4.
3. Line 63: Comma should be replaced by a period.
4. Line 95: How was the exact ratio of the two media decided on?
5. Line 122: Consider already giving the name of the Flavin Reductase at this point.
6. Figure 1: The Y-axis labeling could be more descriptive, as method of GFP-quantification cannot be derived from it. Please elaborate in the caption or change the labeling. Although it is obvious, giving the wavelength used for OD measurement in the graphs would be also preferable.
7. Line 173: The sentence seems unnecessarily complicated. Would it be possible to remove the second instance of "phage enrichment"?
8. Figure 2c: Abbreviation PFU should be explained in the caption. Generally, square brackets would be preferable for units when labeling axes.
9. Line 192: A brief summary on the function of MP6 would help with comprehensibility.
10. Figure 3b: Maybe colour-coding the Y-axes in a similar way to the graphs could improve the immediate readability. Abbreviation PFU should be explained in the caption. X-axis unit should be changed to t [h].
11. Figure 3d: See Figure 1.
12. Figure 4b: In the caption, the 600 in "OD600nm" should be subscript.
13. Line 455: I would suggest consistently using the term "OD600".
14. Line 468: Use multiplication symbol and write g in italic for centrifugation steps.
15. Line 473: Space between 0.75 and μL is missing.

Acceptance is recommended after major revision.

Reviewer #2

(Remarks to the Author)

Reviewer #3

(Remarks to the Author)

In this study the authors design and implement an aminoacyl tRNA synthetase based halogenase biosensor for the PACE-based continuous evolution of soluble and highly active RebH tryptophan

halogenase variants. The best variant, RebHevo4, was used for fermentation-based biomanufacturing and resulted in the production of 2.7 g/L

of halogenated tryptophan, as well as halogenated tryptamines in the range of 200-300 μM , more than 24-35x more efficient compared to wt RebH. Lastly, RebHevo4 was also shown to enable the production of genetically encoded antimicrobial halogenated peptides.

The study is clearly reasoned, and the results presented authoritatively and with conclusions aligned with the presented results.

While PACE has been vastly used for directed evolution campaigns, just as 7-Chl and 7-Br halogenated tryptophan and

tryptamine have been reported in microbial cells previously using wt RebH, the study offers novelty both in terms of novel RebH variants with improved solubility, thermo-stability, and catalytic activity. Similarly, the coupling of the halogenase to the formation of pilli is new, as well as the new antimicrobial peptides.

Taken together the study merits publication should the authors be able to address the following questions and requests.

Major:

Line 202-224 + Suppl. Fig 2: It is unclear how the combinatorial approaches for the generation of evolved RebH variants were instructed. It seems almost stochastic from reading about the procedure used, e.g. "combining different mutations", "tested combinations of all six mutation", and "tested a combination of other mutants". In order to try to better understand the sequence-to-function relationship of the various mutants, the authors should reason their choice of combinatorial mutations as well as systematically compare their e.g. biosensor output performance and degree of solubility, thermo-stability and/or structural changes. This would allow for much better adoption of the evolved variants in the research community (e.g. E coli vs yeast usage; enable model-guided optimisation of future machine-learning guided specificity variants). We are not talking an exhaustive fully-combinatorial variant library, but more a demo of the line of thinking on which mutations to combine.

The study also lacks from more quantitative measures when it comes to the numerous statements on costly/expensive ncAA or toxic/unsustainable chemical processes for halogenation of small molecules and peptides/proteins. The authors should perform a TEA comparing their 5 ltr fermentation process for their 7-Chl-tryptaphan with the process for the chemical synthesis of 7-Chl-tryptaphan. It is acknowledged that the process developed is not commercial, but an indication as to which titer, rates or yields would be needed for the process to be cost-effective. This is particularly relevant since the initial statement in the Abstract on the interest to develop biobased procedures for halogenated products as chemical halogenation requires expensive ingredients.

Minor:

Line 11: The statement on halogenation lacking stereospecificity is too strong, and also irrelevant to the products showcased in this study. Firstly, several halogenation reactions are stereospecific, most notably the anti addition of halogens to alkenes, SN2 halogenation of chiral centers, and halolactonization reactions. Please insert "can lack" or similar. Secondly, the statement falls short as the demonstrated processes in this study are not involving stereochemistry.

Line 103: "and, in effect". Delete?

Line 230 and 234: Do you mean Fig. 3E ins read of 2F?

Reviewer #4

(Remarks to the Author)

This manuscript describes the engineering of a tryptophan halogenase (RebH) using an aaRS-based biosensor in combination with phage-assisted continuous evolution (PACE), resulting in the multi-mutation variant "RebHEvo4" with significantly improved activity in bacterial systems. The authors demonstrate markedly enhanced production levels in whole-cell halotryptophan synthesis at 37 °C compared to the wild type, as well as increased halotryptamine titers when combined with a decarboxylase. They further scale the system to a fed-batch process, achieving 2.7 g/L of 7-chloro-tryptophan, and show that the evolved enzyme enables site-specific halogenation within antimicrobial peptides produced in *E. coli*. In my opinion, this work represents a clear advance over previous halogenase engineering studies, which have largely focused on in vitro activity and stability. The experimental strategy is thoughtfully designed, and the results convincingly demonstrate that continuous evolution can substantially improve enzyme performance in vivo. I particularly appreciate the clear description of the PACE logic, the validation of the biosensor system, and the inclusion of detailed plasmid information, which will aid reproducibility and adoption by the community.

Overall, I recommend publication in Nature Communications, as the manuscript will be of broad interest to readers in synthetic biology, enzyme engineering, and metabolic pathway design. My only suggestion for improvement concerns benchmarking: since the authors reference several prior studies that evolved halogenases through directed evolution, it would be valuable to directly compare RebHEvo4 against a few of these previously reported variants. For example, performing such a comparison in the assay used in Figure 3d, across temperatures such as 30 °C, 34 °C, and 37 °C, would provide an informative context for evaluating the performance gains achieved through PACE. Even if RebHEvo4 is not the top performer, I think such benchmarking would strengthen the manuscript.

Another minor comment: It appears that the following sentence was unintentionally truncated: "We observed that GFP signal was substantially reduced when the media was supplemented with canonical L-tryptophan in addition to 7-Cl-Trp (Figure 1C), indicating that non-halogenated tryptophan acts as an inhibitor of the ChPheRS-4 synthetase and, in effect."

Version 1:

Reviewer comments:

Reviewer #1

(Remarks to the Author)

We thank the authors for appropriately addressing our remarks and expanding the results by incorporating the requested in-vitro assay. The citations were likewise adequately supplemented. We have no further requests regarding changes in the manuscript.

Reviewer #2

(Remarks to the Author)

Reviewer #4

(Remarks to the Author)

I thank the authors for the careful revision and the additional benchmarking experiments. The new data convincingly address my concerns and clearly demonstrate the advantages of RebH Evo4 for in vivo applications. The manuscript is now substantially strengthened, and I support publication in Nature Communications.

Reviewer #1 (Remarks to the Author)

The authors describe in their manuscript a strategy for the continuous evolution of a halogenase, based on the tryptophan-7-halogenase RebH. RebH is a flavin-dependent tryptophan halogenase for which it has been shown that the introduction of a chloro- or bromo-substituent at position 7 of the indole ring of tryptophan is feasible, and that this biocatalysis can also be performed on a gram scale (Angew. Chem. Int. Ed. 2015, 54, 298). Unfortunately, the authors did not cite this work.

Thank you for recommending this paper, we've now cited it in our introduction:

“Previous efforts have managed to reach gram-scale yields using flavin-dependent halogenases in vitro, through an 8-day reaction with a cross-linked aggregate of 3 different purified enzymes (ADH, PrnF, and RebH)¹.

Instead, they state that halogenase biocatalysis has shown only limited success because no substantial amounts of halogenated tryptophans can be produced. This overlooks strategies to obtain halogenated tryptophans either in vitro through cell-free systems (vide supra) or in vivo by metabolic engineering (see Ref. 39).

As a real highlight, the authors report the use of a sensor system based on amber suppression to detect the formation of halogenated tryptophan in vivo. This system relies on a tRNA synthetase that selectively incorporates 7-chlorotryptophan into proteins at TAG codons. The strategy of genetic code extension was combined with the biosynthesis of chlorotryptophan-containing GFP, where fluorescence is observed only upon incorporation of 7-chlorotryptophan. It was also noted that non-halogenated tryptophan acts as an inhibitor of this synthetase.

This GFP readout was then coupled with a continuous phage-assisted evolution (PACE) approach. After 240 hours under varying temperature conditions, thermostable variants of the halogenase were obtained, albeit with only marginal thermostability. Monitoring individual clones over the course of the selection provided further insight into possible substitutions; in addition, error-prone PCR was applied to the selected mutants, and further clonal variants were isolated in a third PACE phase.

This yielded the variant RebH-Evo4, which is several-fold more active than the wild type. It carries 12 mutations, some located at the protein surface, others in the core. At least four mutations are positioned near the flavin-binding site, but none close to the tryptophan-binding site. In this context, it remains an open question whether the regiochemistry of tryptophan halogenation at C7 of the indole ring is affected, even though the mutated residues are not located directly in the tryptophan-binding region. Unfortunately, the work of the Niemann group (J. Biol. Chem. 2019, 294, 2529) on the influence of indole positioning in front of the tunnel between the flavin-binding site and the tryptophan pocket was not cited.

Thanks for flagging this up. We now have included this work in an expanded discussion on modifying halogenase selectivity:

“Notably, none of the twelve adaptive mutations in RebH_{Evo4} fall within its tryptophan binding pocket; an interesting future line of work would combine this with previously reported efforts in engineering RebH to alter its substrate scope. For example, previous efforts have shown it is possible to transplant residues between halogenases to alter their regioselectivity², raising an intriguing possibility that the improvements of RebH_{Evo4} could be applied to synthesis of 5- and

6-halotryptophan simply by transplanting known mutations that confer specificity to those positions³.”

In summary, the authors convincingly demonstrate that a highly active and more soluble halogenase variant could be obtained within a short time by using PACE. They also examined bromination by replacing sodium chloride with sodium bromide, confirming activity, although no data on halide selectivity are presented.

The next section of the manuscript, describing the incorporation of halogenated tryptophan derivatives via genetic code expansion, is a logical continuation of the topic, since the necessary enzymes and methods had already been well established. However, the novelty of this approach is not fully convincing, as the incorporation of non-canonical tryptophan derivatives, including halogenated tryptophans, has already been reported in the literature (see, for example, Kwon & Tirrell, *J. Am. Chem. Soc.* 129, 2007, 10431, and Budisa et al., *Biol. Chem.* 385, 2004, 191).

It has nevertheless to be acknowledged that the selection system for antimicrobial peptides is well chosen.

We appreciate the comment and the acknowledgement of our antimicrobial peptide system. We would like to clarify that we did not mean to claim we are the first ones to add tryptophan analogues into proteins, indeed, the synthetase we use (ChPheRS-4) was tested on halogenated tryptophan incorporation in the publication that created it⁴. Rather, we aimed to use the synthetase/tRNA pair in a novel context as a biosensor and selection mechanism, as well as a bioproduction strategy.

Overall, this is a highly interesting manuscript in the field of enzymatic halogenation, with potential significance for biocatalysis and sustainable bioproduction. The optimization and selection system is well designed, innovative, and highly efficient. The application to ribosomally synthesized antimicrobial peptides is also convincing, even though the incorporation of halogenated tryptophans in proteins using genetic code expansion has been described previously. However, the manuscript suffers somewhat from incomplete citation.

Questions also remain open: for instance, whether surface mutations affect the oligomerization state of RebH, or whether the mutations influence the halogenation regiochemistry and halide selectivity.

In Figure 4, panel a, a slight shift in retention time between products of RebH wild type and RebH-Evo4 is noticeable, which may indicate altered regiochemistry. The problem of product inhibition in enzymatic halogenation by RebH is also not addressed in the discussion.

*Thanks for the comments. Regarding oligomerisation state, halide selectivity, and substrate inhibition, RebH is known to form a stable homodimer, likely stabilised by a crucial salt bridge between Arg387 and Glu432⁵. Obtaining exact values of K_m for Cl⁻ and Br⁻ in FDHs is difficult, since they are likely multiple orders of magnitude lower than the normal biological ion concentration and it is therefore known to be hard to obtain fully salt-free enzyme preparations⁶. As none of our mutations affect this salt bridge, and we observed similar catalytic rates (see new *in vitro* experimental data) on Cl⁻ and Br⁻ ions for both WT RebH and RebH_{Evo4}, we did not pursue experimental characterization of these characteristics. However as we agree changes to them cannot be ruled out, we have added a sentence to the main text clarifying this. Finally, we have also added literature references discussing product inhibition of Flavin-dependent halogenases*

With respect to regiochemistry, we now confirm the evolved enzyme is unchanged in its exclusivity for modification at position C7 with the experiments shown below (Comment 2 from Reviewer 2). We agree that there is a small shift in Figure 4 panel a, however we believe this is actually a result of the methodology we employed for the *in vivo* quantification, where with a UPLC fast run protocol, increasing concentrations of a compound of our *in vivo* extractions can slightly decrease the retention time of the peak, especially at very high concentrations, like those produced by our evolved enzyme. We note that this effect is also observed in the dilution series of purified standards, as well as for the tryptophan peak (see figure below), and therefore do not believe it compromises the assay's ability to quantify product titre, especially since the UPLC was coupled with a downstream MS analysis, which verified the masses of all peaks used for quantification.

We have added updated the relevant section in the main text “Evolved RebH_{Evo4} is more soluble and more active than the WT” with these changes:

“Indeed, the evolved enzyme showed improved catalysis, with a ~2.5-fold increase in apparent k_{cat} at saturating tryptophan concentrations^{6,7} for chlorination at both temperatures, and similar improvements for bromination (Supplementary Fig. 5). The regioselectivity of the evolved enzyme did not change, still exclusively halogenating tryptophan at the C7 position (Supplementary Fig. 6). Finally, we do note that there are other aspects of RebH function we did not test which could have changed as well, such as better dimerisation or reduced substrate inhibition, which are both known to affect halogenase activity^{8,9}.”

In Figure 5g, determination of the exact mass of the respective peptides would have been desirable.

The expected mass ($MW_{Exp.}$) and exact observed mass ($MW_{Obs.}$) for all peptides and proteins is displayed in the figure and are all within 0.5 Da of their expected values, which is very small and an accepted level of variation for whole-protein analysis.

Finally, the shoulder in the HPLC analysis (Supplementary Figure 6g) of the halogenated enterocin RJ11 derivative also requires explanation.

We analysed the samples on an Agilent 1260 LC-MSD HPLC/MS instrument, which was configured for use with high-concentration solid-phase peptide synthesis samples, and consequently used a saved autoblank as a reference for each run. This means background signal can be somewhat higher than with other machines, causing such a shoulder to appear on samples run at lower concentrations like the purified, bioproduced peptide.

The presented work utilizes a sophisticated evolution strategy to achieve impressive increases in productivity and solubility of the halogenase RebH. Overall, the results are very well presented and supported with the data shown. Apart from two comments regarding additional data, I only have minor comments:

1. An *in vitro* comparison of the new variant RebHEvo4 to the respective wild type would help to further validate the claimed improvement in activity and stability. Based on the *in vivo* assay, it is not entirely clear if the increase in product formation can be solely attributed to a better solubility in the cellular environment or if an increased activity of the enzyme contributes to this. Although you make a compelling case for the advantages of using halogenases *in vivo*, it would be interesting to see if the observed advantages of this new mutant translate to smaller scale *in vitro* use.

We thank the reviewer for this comment. We agree that whether the improvements we observe in RebH_{Evo4} come solely from its improved solubility or are also a product of increased catalytic activity is an important question, and would improve the manuscript. We therefore decided to pursue *in vitro* measurements for the catalytic rate in chlorination and bromination by WT and evolved RebH. To make our results as reproducible and comparable to previous reports as possible, we decided to follow a very similar protocol to that used in Payne et al., 2013. Following expression and purification of RebH variants and their partner flavin reductase RebF, we measured the catalytic rate in both chlorination and bromination at saturating concentrations of all substrates. For both halides, our evolved enzyme showed an approximately 2.5 fold increase in apparent k_{cat} , indicating that the improved *in vivo* activity of RebH_{Evo4} comes from changes not only to solubility, but also to catalysis.

We have added a new Supplementary Figure (5), an extensive new section in the Methods, and refer to these added results in the following added paragraph in the main text:

*“To evaluate whether the activity increase observed for RebH_{Evo4} is solely attributable to improved solubility, or whether its catalysis may also have been enhanced by evolution, we used an established *in vitro* assay⁷ to compare the catalytic rate of purified RebH_{Evo4} with RebH_{WT} at 30 °C and 37 °C. Indeed, the evolved enzyme showed improved catalysis, with a ~2.5-fold increase in apparent k_{cat} at saturating tryptophan concentrations^{6,7} for chlorination at both temperatures, and similar improvements for bromination (Supplementary Fig. 5).”*

$n = 6$ for chlorination at 37°C (on two different days with two different mastermix batches)
 $n = 3$ for all other assays (three independent reactions on same day)

Data represents mean initial conversion velocities of enzyme reactions performed in triplicate (6 replicates for chlorination at 37°C), with 150 μM L-Trp and 10 mM NaCl (chlorination) or 100 mM NaBr (bromination), which have previously been reported as saturating substrate concentrations [Andorfer et al. 2013]. Enzyme-assays were performed under air-saturated conditions with vigorous shaking (see Methods).

2. Even if the binding site is not affected, a total of 12 mutations could potentially affect the global fold and therefore even substrate binding. Could incorporation of e.g. C5- or C6-halotryptophans by ChPheRS-4 be ruled out entirely in this case? If not, NMR-analysis of the halogenated products would be beneficial to confirm the regioselectivity of RebHEvo4.

The reviewer makes a valid point, with the data currently reported we can't say for sure if regioselectivity of RebH_{Evo4} has changed. We know from past experiments that ChPheRS-4 does not tolerate C5-modified tryptophans, however it is possible for it to promiscuously incorporate C6-halotryptophan (albeit at low efficiency). In order to address whether this could have resulted in reduced regioselectivity in RebH_{Evo4}, we performed an experiment to see if the product of the two enzymes showed any difference in the concentrations of C5, C6 and C7-halotryptophans produced. It has been previously reported that HPLC is sufficient to quantify these three regioisomers of halotryptophan^{3,10}, and after successfully finding a satisfactory method in which we saw strong separation between the three, we ran samples of the halogenated product (taken from the *in vitro* experiment mentioned above) produced by both RebH_{WT} and RebH_{Evo4}. The results confirmed, via retention time, that both the WT and evolved enzymes produce exclusively C7-halotryptophans. This data has now been added to the manuscript as Supplementary Fig. 6.

3. Line 63: Comma should be replaced by a period.

Thank you, this has been corrected.

4. Line 95: How was the exact ratio of the two media decided on?

DRM is the optimised media generally used for PACE, however we observed that in terms of raw circuit performance, M9 gave higher signal on our GFP readout (Figure 1B). To ensure we maintained high circuit performance without compromising phage activity, a 1:9 and 5:95 ratio of the two media was tested initially, and both already gave good fold-change in signal, with good growth. Since we obtained robust growth at 1:9, we decided to move on with this ratio, and didn't test it further. We have now added this information in the methods section.

5. Line 122: Consider already giving the name of the Flavin Reductase at this point.

Thank you, this has been corrected.

6. Figure 1: The Y-axis labeling could be more descriptive, as method of GFP-quantification cannot be derived from it. Please elaborate in the caption or change the labeling. Although it is obvious, giving the wavelength used for OD measurement in the graphs would be also preferable.

All GFP measurements in this manuscript are reported as GFP fluorescence normalised by bacterial growth as measured using OD_{600} . This ensures that changes in growth rate do not impact the fidelity of measurements. GFP/OD is a standard term for this measurement¹¹, however to further clarify the point for readers who may be familiar we have added a line in our methods section and a short sentence in captions. Also, OD_{600} was added to all captions.

7. Line 173: The sentence seems unnecessarily complicated. Would it be possible to remove the second instance of "phage enrichment"?

Thank you, this has been corrected, the sentence has been revised to increase readability.

8. Figure 2c: Abbreviation PFU should be explained in the caption. Generally, square brackets would be preferable for units when labeling axes

We have added an explanation of the term “PFU/mL. All figure axis labels have been changed to use square brackets”.

9. Line 192: A brief summary on the function of MP6 would help with comprehensibility.

10. Figure 3b: Maybe colour-coding the Y-axes in a similar way to the graphs could improve the immediate readability. Abbreviation PFU should be explained in the caption. X-axis unit should be changed to t [h].

11. Figure 3d: See Figure 1.

12. Figure 4b: In the caption, the 600 in “OD600nm” should be subscript.

13. Line 455: I would suggest consistently using the term “OD600”.

14. Line 468: Use multiplication symbol and write g in italic for centrifugation steps.

15. Line 473: Space between 0.75 and μL is missing.

Thank you so much for catching these typos and suggestions to the text. They have been corrected in the revised manuscript. We did not change the colour of the Y-axes on figure 3b though, as we found changing it to blue more difficult to read.

Acceptance is recommended after major revision.

Reviewer #2 (Remarks to the Author)

Reviewer #3 (Remarks to the Author)

In this study the authors design and implement an aminoacyl tRNA synthetase based halogenase biosensor for the PACE-based continuous evolution of soluble and highly active RebH tryptophan halogenase variants. The best variant, RebHevo4, was used for fermentation-based biomanufacturing and resulted in the production of 2.7 g/L of halogenated tryptophan, as well as halogenated tryptamines in the range of 200-300 μ M, more than 24-35x more efficient compared to wt RebH. Lastly, RebHevo4 was also shown to enable the production of genetically encoded antimicrobial halogenated peptides.

The study is clearly reasoned, and the results presented authoritatively and with conclusions aligned with the presented results. While PACE has been vastly used for directed evolution campaigns, just as 7-Chl and 7-Br halogenated tryptophan and tryptamine have been reported in microbial cells previously using wt RebH, the study offers novelty both in terms of novel RebH variants with improved solubility, thermo-stability, and catalytic activity. Similarly, the coupling of the halogenase to the formation of pilli is new, as well as the new antimicrobial peptides.

Taken together the study merits publication should the authors be able to address the following questions and requests.

Major:

Line 202-224 + Suppl. Fig 2: It is unclear how the combinatorial approaches for the generation of evolved RebH variants were instructed. It seems almost stochastic from reading about the procedure used, e.g. “combining different mutations”, “tested combinations of all six mutations”, and “tested a combination of other mutants”. In order to try to better understand the sequence-to-function relationship of the various mutants, the authors should reason their choice of combinatorial mutations as well as systematically compare their e.g. biosensor output performance and degree of solubility, thermo-stability and/or structural changes. This would allow for much better adoption of the evolved variants in the research community (e.g. E coli vs yeast usage; enable model-guided optimisation of future machine-learning guided specificity variants). We are not talking about an exhaustive fully-combinatorial variant library, but more a demo of the line of thinking on which mutations to combine.

We appreciate the reviewer for raising this point. We have thoroughly reviewed our manuscript and considered how to better communicate our rationale for choosing certain mutants or mutagenesis approaches throughout the manuscript. We have now updated it with several changes in the section “Halogenation-dependent phage propagation”.

As an overview, the main rationale for selecting which mutations to combine followed the same order of priority:

- I. Fixed mutations in the phage population (more than 50% of phages had those mutations) were always selected as the best variants and the “starting point” for the additional combinations.
- II. Enriched phages, with more than 2 sequenced phages containing that particular mutation were then prioritized for exploration, and always all tested (either alone or in combination).
- III. Mutations found in single plaque. For these mutations, the priority for testing was:
 - a. co-occurrence with either a fixed mutation or enriched mutation (for example, T348A co-occurred with V256I and T385I in a phage),

- b. occurrence at close proximity of other observed mutations (For example, D101N and G102S, and T496R and Q494K, G504S and R509C), which could suggest a mutation hot spot,
- c. close proximity to catalytic sites of the protein (For example, N326K and L233M) or surface residues that add charge (such as Q323H, E186K, Q494K). Few mutations, like V481I, were tested due to its presence in regions of the protein structure that had no mutations close to it.

We could not, however, test all the possible mutations from category C, due to the limitation on the number of mutations we tested or could combine (ordering gene blocks and cloning using oligos) and we acknowledge that there might be valuable combinations and residues not explored, However, we believe that we tested a satisfactory number of combinatorial mutants (29 in total) to allow us to obtain a highly efficient enzyme, which at one point we decided to carry on to the full characterization panel (in-vivo activity, solubility and now, in-vitro experiment).

Finally, it is known that during continuous or sequential evolution campaigns, a lot of the observed mutations are epistatic, therefore we believe that characterizing them individually and trying to assess their contribution to every aspect is not the most efficient way to report these improvements. In fact, we observed this with our data, as exemplified below; Testing a single phage mutant obtained during the first PACE round (T385I) versus the double mutant (T385I - V256I, which dominated the phage population) show a clear benefit from the double mutant. However, it very quickly becomes obvious that mutations can be epistatic when just by testing a few more mutations obtained in the first round. M430L seems to have a detrimental effect to the double mutant, but when combined with T348A, it does provide an increase in activity:

Similarly, as observed with A50T and A16S mutations (Supp. Figure 2D) shows a clear detrimental effect of combining both mutations, even though both mutations were selected during PACE and enriched.

However, we do agree with the reviewer that having raw data from a combination of different mutations is valuable for future machine learning-guided protein engineering efforts. Considering we tested a significant number of different combinations (in total 29), we decided to include a table of all raw data for fold-change (versus WT) and GFP/OD for all the 29 different combinations of mutations tested in this manuscript. This data can now be found as the new Supplementary Table 2. We would like to thank the reviewer for pointing these issues out, since it has made the manuscript clearer.

The study also lacks from more quantitative measures when it comes to the numerous statements on costly/expensive ncAA or toxic/unsustainable chemical processes for halogenation of small molecules and peptides/proteins. The authors should perform a TEA comparing their 5 ltr fermentation process for their 7-Chl-tryptaphan with the process for the chemical synthesis of 7-Chl-tryptaphan. It is acknowledged that the process developed is not commercial, but an indication as to which titer, rates or yields would be needed for the process to be cost-effective. This is particularly relevant since the initial statement in the Abstract on the interest to develop biobased procedures for halogenated products as chemical halogenation requires expensive ingredients.

Thank you for this comment. We agree that these are important questions, and worked on a TEA-style investigation in order to better substantiate the claims we make about cost in our manuscript. In general, one of the big challenges with undertaking a full TEA outside of halogenated small molecule or peptide manufacturing firms is the paucity of information publicly-available about the real cost of commercial 7-CI-Tryptophan or bioactive peptide synthesis. Additionally, to perform an in-depth TEA there are several required values that are very difficult to estimate just by analysing results from a pilot run, such as bulk orders for reagents, plant life, depreciation schedule, market size, to mention a few¹². That being said, it is possible to work out some estimations, and we aimed to give an impression of costs and feasibility for halogenated 7-CI-Trp/7-Br-Trp and halogenated peptides:

Currently, Tryptophan is produced industrially using microbial fermentation with modified *E. coli* or *C. glutamicum* strains (by companies like Ajinomoto and CJ Bio), with yields expected to be between 40g/L to 50g/L per run^{13,14}. Tryptophan can be purchased at different purity grades from between 200\$/kg to 1200\$/kg, it is also used as a food additive and for animal nutrition. The demand for halogenated tryptophan is, of course, much lower than for regular tryptophan, since it is used only in highly specialized products and research. If we don't consider market demands and just focus on current costs, halogenated Trp molecules have much higher value, with average costs ranging from \$200 to \$1,000 a gram.

The media run cost using an M9-like media ranges around \$5-\$10 per 5L¹⁵. Adding 20g per 5L of Tryptophan would cost around \$4. If we add additional costs, such as, inducers (IPTG), electricity and heating¹⁶ and depreciation of the bioreactor, this can add around \$50 per run, but could decrease if scaled up. Higher purity of the final material might be needed (potentially using an ion exchange column), which will incur extra costs that can account up to 40% of production costs of bulky biochemicals^{17,18}, therefore in the range of \$50 per 5L. In total, a rough estimation, the total cost per 5L bioreactor run would be close to \$120 for 13.5g of 7-Halo-Trp. If we consider a 30-50% loss during purification, this would still account for around 7g to 9g of final material, which would be worth close to \$1,400-\$3,000 in commercial value.

Another estimation for feasibility could be done comparing it with existing industrial processes: considering that the price of halotryptophans are about 100x higher than regular tryptophan, it could be expected that even a 100x drop in fermentation yields, compared to tryptophan fermentation (average 45g/L), would still make such a product viable, if we estimate that the production costs for both products are similar. Our observed yields are 2.7g/L, only one magnitude less than for tryptophan, so could potentially be viable, assuming that such yields can be kept on a large scale. Finally, in our process, we obtained a total of 13.5 grams using a 5L bioreactor, but we believe that final yield could be improved by optimizing fermentation conditions and using stronger promoters to drive Fre/RebH, such as T7.

For the halogenated peptide, we could estimate a similar cost for the fermentation run (around \$120 per 5L). The commercial price asked by companies to synthesize peptides (produced using SPPS, small scale (up to 50mg) with >95% purity)) with 40aa (size of Enterocin RJ-11) is around \$50/mg, but can go up to \$150/mg when considering the addition of non-canonical amino-acids. Therefore, in order to be cost-comparable with commercial prices, considering \$120 per 5L run, peptide yields after purification would need to be 1mg to 2mg per 5L. This is well within the range of previously described peptides expressed using *E. coli*, which can range from 25mg/L to 40mg/L of final, purified product^{19,20}. Importantly, as any industrial process, cost of recombinant protein/peptides can decrease as much as 80% when scaled up, and have already been shown to be cost competitive versus chemical synthesis by other groups¹⁹. Finally, SPPS for peptide production is known to be an unsustainable process, which uses large amounts of solvents (DCM, DMF) and generates a lot of waste^{21,22}. This has now been added to the manuscript.

As stated initially, TEAs are complex calculations and real, verifiable, industrial-scale production costs are challenging since this information is proprietary and closely guarded by companies. Committing to hard numbers and an in-depth TEA is a difficult task, requiring estimation of bulk order cost, waste management, plant life, depreciation schedule, among many other factors, and such efforts normally constitute whole manuscripts on their own (see^{23,24}). We sincerely believe that a full TEA is beyond the scope of this manuscript, however, we do agree with the reviewer that it could benefit from more information about the costs associated with chemical production of peptides and halogenated amino acids. We have addressed this by adjusting the language in our abstract and adding the following addition to the discussion to give the reader more context for cost, yield, and solvent use.

“While solid-phase peptide synthesis (SPPS) is commonly used, it becomes inefficient for long peptides or proteins and is environmentally-unfriendly due to high volumes of solvent and waste²¹. In contrast, biological peptide synthesis has been demonstrated to be cost competitive, and avoids these environmental concerns. For example, Gaglione et al. demonstrate that in an optimized system biological production costs can be reduced down to 42 €/mg of final peptide, around 10x less than commercial prices for chemical synthesis strategies¹⁹

*Industrial production of tryptophan itself is routinely performed in modified *E. coli* and *C. glutamicum*, reaching titres in the range of 40-50 g/L^{13,14}, and its halogenated counterparts are often around two to three orders of magnitude more valuable. Our final titre in a 5L bioreactor of 2.7 g/L is therefore encouraging, especially considering that it is likely yields could be further improved by optimizing fermentation conditions and using stronger expression to drive halogenation machinery.”*

Minor:

Line 11: The statement on halogenation lacking stereospecificity is too strong, and also irrelevant to the products showcased in this study. Firstly, several halogenation reactions are stereospecific, most notably the anti addition of halogens to alkenes, SN2 halogenation of chiral centers, and halolactonization reactions. Please insert “can lack” or similar. Secondly, the statement falls short as the demonstrated processes in this study are not involving stereochemistry.

Thanks for this comment and flagging this up. We have revised the sentence and added “can lack”.

Line 103: “and, in effect”. Delete?

Line 230 and 234: Do you mean Fig. 3E ins read of 2F?

We thank the reviewer for catching these typos and have corrected them all in the text.

Reviewer #4 (Remarks to the Author)

This manuscript describes the engineering of a tryptophan halogenase (RebH) using an aaRS-based biosensor in combination with phage-assisted continuous evolution (PACE), resulting in the multi-mutation variant “RebHEvo4” with significantly improved activity in bacterial systems. The authors demonstrate markedly enhanced production levels in whole-cell halotryptophan synthesis at 37 °C compared to the wild type, as well as increased halotryptamine titers when combined with a decarboxylase. They further scale the system to a fed-batch process, achieving 2.7 g/L of 7-chloro-tryptophan, and show that the evolved enzyme enables site-specific halogenation within antimicrobial peptides produced in *E. coli*.

In my opinion, this work represents a clear advance over previous halogenase engineering studies, which have largely focused on *in vitro* activity and stability. The experimental strategy is thoughtfully designed, and the results convincingly demonstrate that continuous evolution can substantially improve enzyme performance *in vivo*. I particularly appreciate the clear description of the PACE logic, the validation of the biosensor system, and the inclusion of detailed plasmid information, which will aid reproducibility and adoption by the community.

Overall, I recommend publication in Nature Communications, as the manuscript will be of broad interest to readers in synthetic biology, enzyme engineering, and metabolic pathway design.

My only suggestion for improvement concerns benchmarking: since the authors reference several prior studies that evolved halogenases through directed evolution, it would be valuable to directly compare RebHEvo4 against a few of these previously reported variants. For example, performing such a comparison in the assay used in Figure 3d, across temperatures such as 30 °C, 34 °C, and 37 °C, would provide an informative context for evaluating the performance gains achieved through PACE. Even if RebHEvo4 is not the top performer, I think such benchmarking would strengthen the manuscript.

We agree with the reviewer that benchmarking our evolved RebH against other described halogenases is important to help readers contextualise the utility of our evolved enzyme, and would strengthen the manuscript. We therefore decided to test RebH_{Evo4} against other reported improved tryptophan-7 halogenase variants *in vivo* using our aaRS biosensor with GFP as a readout.

It is important to note that while we do mention papers reporting engineered halogenases, many of these actually refer to halogenases with different regiospecificity (such as Thal²⁵, which is a tryptophan-6 halogenase), and so cannot be directly benchmarked against our results. Furthermore, in the context of engineered tryptophan-7 halogenase variants (including RebH), most of these efforts actually focus on mutagenising the binding pocket or otherwise engineering the enzyme in the hope of recapitulating WT-level activity for novel substrates such as tryptamines, carbazoles, and modified indoles^{26,27}. Our goal was to improve RebH activity overall and push beyond the WT by addressing underlying problems with the enzyme like poor solubility, activity and thermostability, therefore comparing these halogenase variants with RebH_{Evo4} would not be particularly meaningful. There has been, however, one effort by the Lewis lab to improve RebH in a similar manner, in which they evolve the enzyme with the intention of making it more active and thermostable, but for *in vitro* use²⁸. They report two main variants; 3-LR and 3-LSR with a reported increase in activity towards tryptophan (up to four-fold increase in conversion) and increase in optimum temperature from 35 °C to 40 °C. To our best

knowledge, these represent the only engineered tryptophan-7 halogenase variants currently reported as having significantly increased overall enzyme activity and properties.

We first started by transplanting the mutations of the 3-LR and 3-LSR variants into the protein coding sequence of WT RebH in our expression construct, pAP65. This ensured that — to the extent possible — effects like codon optimization could not obscure differences in actual protein-level properties. Poor et al. report that the optimum temperature of their evolved enzymes are both close to their maximum between 35-40 degrees, with a fall-off below 35 degrees and above 40 degrees, so we ran the assay at both 34 and 37. Unfortunately, both of these variants displayed very little activity *in vivo* at both temperatures. To be sure that the low signal observed was due to weak halogenase activity, we repeated the assay, supplementing the variants with 200 μ M 7-CI-Trp. We observed strong GFP signal, confirming that there were no issues with the genetic circuit.

We did note that Poor and collaborators evolved these variants in the context of a 20 amino acid N-terminal His-Thrombin tag, and they bear an S2P mutation in very close proximity to it. To be absolutely sure that the results we saw were not due to variants having evolved dependence on this tag's presence, we subsequently cloned the exact full polypeptide sequence of their 3-LR and 3-LSR variants, including the tag, as reported in the PDB entry for 3-LSR (4LU6). We also switched from NEB 10-beta to our Δ tnaA strain sAP50, since we know it can improve performance of low-activity variants. However, even these changes did not ameliorate the low *in vivo* activity levels we observed, with all variants showing only barely detectable signal above a no-halogenase control used to estimate background

It is an unfortunate but well-demonstrated problem that improvements gained from evolving proteins *in vitro* can fail to translate to *in vivo* use, and can occasionally make them worse by increasing propensity for toxicity or aggregation. We believe that the effects we observe stem from such a case, where these variants have evolved to be capable of performing extremely well *in vitro* but have in doing so acquired mutations that make them incompatible with *in vivo* use. The selection employed by Poor et al. differs quite drastically from the conditions of an actual living cell, occurring in cell lysate after a significant high temperature heat-shock step, and so would not necessarily result in mutants which are still capable of behaving well in a complex living environment. We note that in the 11 years since 3-LR and 3-LSR were first reported, despite the many papers attempting to use RebH *in vivo*, WT RebH sequences remain the rule^{29–33}. Fundamentally, it seems that the most effective way to evolve proteins for *in vivo* use is through *in vivo* evolution.

The final benchmarking experimental data has now been added to the manuscript as Supp. Figure 7.

Another minor comment: It appears that the following sentence was unintentionally truncated: “We observed that GFP signal was substantially reduced when the media was supplemented with canonical L-tryptophan in addition to 7-Cl-Trp (Figure 1C), indicating that non-halogenated tryptophan acts as an inhibitor of the ChPheRS-4 synthetase and, in effect.”

Thank you so much for catching these typos, they have been corrected in the revised manuscript.

References

1. Frese, M. & Sewald, N. Enzymatic halogenation of tryptophan on a gram scale. *Angew. Chem. Int. Ed.* **54**, 298–301 (2015).
2. Moritzer, A.-C. *et al.* Structure-based switch of regioselectivity in the flavin-dependent tryptophan 6-halogenase Thal. *J. Biol. Chem.* **294**, 2529–2542 (2019).
3. Andorfer, M. C., Park, H. J., Vergara-Coll, J. & Lewis, J. C. Directed evolution of RebH for catalyst-controlled halogenation of indole C-H bonds. *Chem. Sci.* **7**, 3720–3729 (2016).
4. Ding, W. *et al.* Chimeric design of pyrrolysyl-tRNA synthetase/tRNA pairs and canonical synthetase/tRNA pairs for genetic code expansion. *Nat. Commun.* **11**, 3154 (2020).
5. Bitto, E. *et al.* The structure of flavin-dependent tryptophan 7-halogenase RebH. *Proteins* **70**, 289–293 (2008).
6. Yeh, E., Garneau, S. & Walsh, C. T. Robust in vitro activity of RebF and RebH, a two-component reductase/halogenase, generating 7-chlorotryptophan during rebeccamycin biosynthesis. *Proc. Natl. Acad. Sci. U. S. A.* **102**, 3960–3965 (2005).
7. Payne, J. T., Andorfer, M. C. & Lewis, J. C. Regioselective Arene Halogenation using the FAD-Dependent Halogenase RebH. *Angew. Chem. Weinheim Bergstr. Ger.* **125**, 5379–5382 (2013).
8. Ismail, M. *et al.* Straightforward regeneration of reduced flavin adenine dinucleotide required for enzymatic tryptophan halogenation. *ACS Catal.* **9**, 1389–1395 (2019).
9. Besse, C., Niemann, H. H. & Sewald, N. Increasing the stability of flavin-dependent halogenases by disulfide engineering. *Chembiochem* **25**, e202300700 (2024).
10. Sher, H. *et al.* SsDiHal: discovery and engineering of a novel tryptophan dihalogenase. *J. Biol. Eng.* **19**, 59 (2025).
11. d'Oelsnitz, S. *et al.* Using fungible biosensors to evolve improved alkaloid biosyntheses. *Nat. Chem. Biol.* **18**, 981–989 (2022).
12. Poddar, T. K. & Scown, C. D. Technoeconomic analysis for near-term scale-up of bioprocesses. *Curr. Opin. Biotechnol.* **92**, 103258 (2025).

13. Liu, L., Duan, X. & Wu, J. L-tryptophan production in *Escherichia coli* improved by weakening the PTA-AckA pathway. *PLoS One* **11**, e0158200 (2016).
14. Ikeda, M. & Katsumata, R. Hyperproduction of tryptophan by *Corynebacterium glutamicum* with the modified pentose phosphate pathway. *Appl. Environ. Microbiol.* **65**, 2497–2502 (1999).
15. Kumar Gupta, P. & Reddy Edula, J. Strategies for enhancing product yield: Design of experiments (DOE) for *Escherichia coli* cultivation. in *Fermentation - Processes, Benefits and Risks* (IntechOpen, 2021).
16. Cardoso, V. M. *et al.* Cost analysis based on bioreactor cultivation conditions: Production of a soluble recombinant protein using *Escherichia coli* BL21(DE3). *Biotechnol. Rep. (Amst.)* **26**, e00441 (2020).
17. Straathof, A. J. J. The proportion of downstream costs in fermentative production processes. in *Comprehensive Biotechnology* 811–814 (Elsevier, 2011).
18. Janković, T., Straathof, A. J. J. & Kiss, A. A. Eco-efficient downstream processing of 1,3-propanediol applicable to various fermentation processes. *Process Biochem.* **143**, 210–224 (2024).
19. Gaglione, R. *et al.* Cost-effective production of recombinant peptides in *Escherichia coli*. *N. Biotechnol.* **51**, 39–48 (2019).
20. Gomez-Lugo, J. J., Casillas-Vega, N. G., Gomez-Loredo, A., Balderas-Renteria, I. & Zarate, X. High-yield expression and purification of scygonadin, an antimicrobial peptide, using the small metal-binding protein SmbP. *Microorganisms* **12**, 278 (2024).
21. Isidro-Llobet, A. *et al.* Sustainability challenges in peptide synthesis and purification: From R&D to production. *J. Org. Chem.* **84**, 4615–4628 (2019).
22. Mthethwa, N. *et al.* Toward sustainable solid-phase peptide synthesis strategy – *in situ* Fmoc removal. *Green Chem. Lett. Rev.* **17**, (2024).
23. Ferreira, R. da G., Azzoni, A. R. & Freitas, S. Techno-economic analysis of the industrial production of a low-cost enzyme using *E. coli*: the case of recombinant β -glucosidase. *Biotechnol. Biofuels* **11**, (2018).

24. Akmayan, I., Ozturk, A. B. & Ozbek, T. Recombinant proteins production in Escherichia coli BL21 for vaccine applications: a cost estimation of potential industrial-scale production scenarios. *Prep. Biochem. Biotechnol.* **54**, 932–945 (2024).
25. Prakinee, K. *et al.* Mechanism-guided tunnel engineering to increase the efficiency of a flavin-dependent halogenase. *Nat. Catal.* **5**, 534–544 (2022).
26. Sana, B. *et al.* Engineered RebH Halogenase Variants Demonstrating a Specificity Switch from Tryptophan towards Novel Indole Compounds. *Chembiochem* **22**, 2791–2798 (2021).
27. Payne, J. T., Poor, C. B. & Lewis, J. C. Directed evolution of RebH for site-selective halogenation of large biologically active molecules. *Angew Chem Int Ed Engl* **54**, 4226–4230 (2015).
28. Poor, C. B., Andorfer, M. C. & Lewis, J. C. Improving the stability and catalyst lifetime of the halogenase RebH by directed evolution. *Chembiochem* **15**, 1286–1289 (2014).
29. Lee, J. *et al.* Production of Tyrian purple indigoid dye from tryptophan in Escherichia coli. *Nat Chem Biol* **17**, 104–112 (2021).
30. Reed, K. B. *et al.* A modular and synthetic biosynthesis platform for de novo production of diverse halogenated tryptophan-derived molecules. *Nat Commun* **15**, 3188 (2024).
31. Milne, N. *et al.* Engineering Saccharomyces cerevisiae for the de novo Production of Halogenated Tryptophan and Tryptamine Derivatives. *ChemistryOpen* **12**, e202200266 (2023).
32. Bradley, S. A. *et al.* Biosynthesis of natural and halogenated plant monoterpene indole alkaloids in yeast. *Nat Chem Biol* **19**, 1551–1560 (2023).
33. Lai, H.-E. *et al.* GenoChemetic Strategy for Derivatization of the Violacein Natural Product Scaffold. *ACS Chem Biol* **16**, 2116–2123 (2021).